# AlpacaFarm: A Simulation Framework for Methods that Learn from Human Feedback

**Yann Dubois**[*]
Stanford

**Xuechen Li**[*]
Stanford

**Rohan Taori**[*]
Stanford

**Tianyi Zhang**[*]
Stanford

**Ishaan Gulrajani**
Stanford

**Jimmy Ba**
University of Toronto

**Carlos Guestrin**
Stanford

**Percy Liang**
Stanford

**Tatsunori B. Hashimoto**
Stanford

## Abstract

Large language models (LLMs) such as ChatGPT have seen widespread adoption due to their ability to follow user instructions well. Developing these LLMs involves a complex yet poorly understood workflow requiring training with human feedback. Replicating and understanding this instruction-following process faces three major challenges: the high cost of data collection, the lack of trustworthy evaluation, and the absence of reference method implementations. We address these challenges with AlpacaFarm, a simulator that enables research and development for learning from feedback at a low cost. First, we design LLM prompts to simulate human feedback that are 45x cheaper than crowd-workers and display high agreement with humans. Second, we propose an automatic evaluation and validate it against human instructions obtained on real-world interactions. Third, we contribute reference implementations for several methods (PPO, best-of-$n$, expert iteration, and more) that learn from pairwise feedback. Finally, as an end-to-end validation of AlpacaFarm, we train and evaluate eleven models on 10k pairs of real human feedback and show that the rankings of models trained in AlpacaFarm match the rankings of models trained on human data. As a demonstration of the research possible in AlpacaFarm, we find that methods that use a reward model can substantially improve over supervised fine-tuning and that our reference PPO implementation leads to a +10% improvement in win-rate against Davinci003. We release AlpacaFarm at https://github.com/tatsu-lab/alpaca_farm.

## 1 Introduction

Large language models (LLMs) [10, 13, 48] have demonstrated unprecedented capabilities in following diverse and open-ended instructions [49, 5, 40]. These achievements have often been attributed to the fine-tuning of pretrained LLMs using human feedback, but this process remains poorly understood due to the lack of published information on the training methods from LLM vendors. For example, it was recently revealed that only the Davinci003 model in the instruct series of OpenAI models used reinforcement learning (RL) with the PPO algorithm [47], leading some to question the importance of RL in the training process. Understanding and improving these methods requires open and transparent replications of the training process, but this remains challenging due to the cost and complexity associated with methods that learn from human feedback.

Our goal is to facilitate research and development on instruction following models and methods that learn from human feedback. We identify three main challenges: the high cost of data annotation, the lack of automated evaluation for model development, and the absence of validated implementations

---

[*]Equal contribution; random order. Contact {yanndubs, lxuechen, rtaori, tz58}@stanford.edu.

37th Conference on Neural Information Processing Systems (NeurIPS 2023).

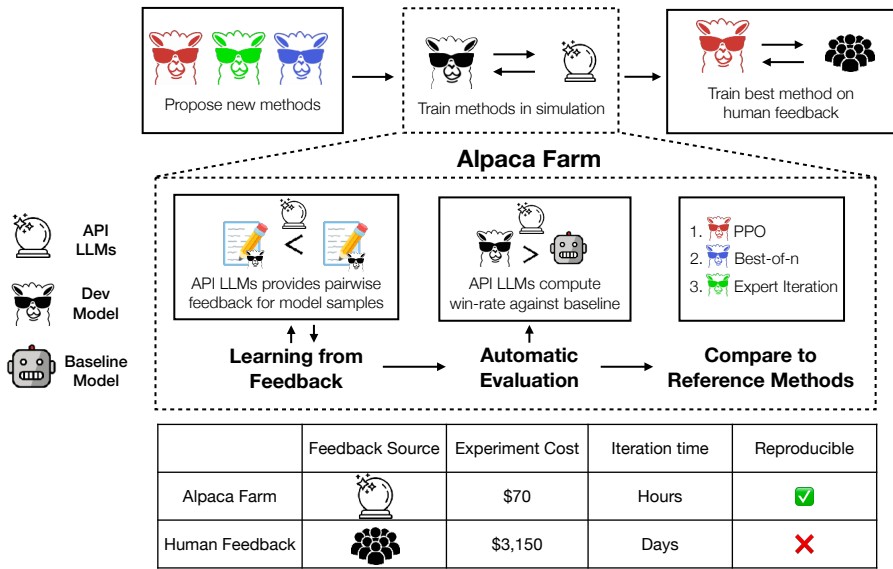

Figure 1: AlpacaFarm is a simulation sandbox that enables fast and cheap experimentation on methods that learn from human feedback. It simulates human feedback with API LLMs, provides a validated evaluation protocol, and offers a suite of reference method implementations. Researchers can iterate on model development and transfer their methods to training on human data to maximize performance.

of existing methods. To address these three challenges, we introduce AlpacaFarm (Figure 1), a simulation sandbox that enables experimentation at a low cost. Using AlpacaFarm, researchers can rapidly iterate on method development in simulation and transfer these insights to build high-performance systems with actual human feedback.

For the first challenge of data annotation costs, AlpacaFarm simulates human annotators with API LLMs that are faster and lower cost. To collect simulated feedback data, we design prompts for API LLMs (e.g. GPT-4) that enable us to simulate human pairwise comparisons at a cost that is 45x cheaper than crowd-workers, and tune these prompts to faithfully capture many aspects of human annotators, such as their quality judgments, inter-annotator variability, and stylistic preferences.

For the second challenge of automated evaluation, we design an automatic evaluation protocol that aims to quantify system performance on simple but realistic real-world human instructions. Improving evaluations of open-ended instruction following has been challenging due to the cost and non-replicability of human evaluation, the lack of real human interaction data, and the diversity of natural human instructions. To address this, we use instructions from user interactions (Alpaca Demo [69]) as a reference for simple but real human interactions and show that we can combine existing evaluation datasets to mimic this evaluation. Quantitative evaluations of system rankings on our evaluation data show a high correlation with system rankings on the Alpaca Demo instructions.

For the third challenge of missing reference implementations, we implement and test several popular learning algorithms including PPO [62], expert iteration [2], and Quark [42], and release reference implementations. We show that, among the methods we studied, PPO with a surrogate reward model is the most effective training-time method in our leaderboard, improving the win-rate against Davinci003 of an instruction fine-tuned LLaMA 7B model from 44% to 55%. Other baselines that have been validated on simpler tasks fall short in comparison, highlighting the importance of testing these algorithms in a real instruction-following setting.

As an end-to-end evaluation of the AlpacaFarm, we compare eleven methods trained and evaluated in AlpacaFarm with the same methods trained and evaluated on actual human feedback. We show that the method rankings obtained from developing on AlpacaFarm closely agree with the method rankings obtained from training on actual human data (Spearman correlation of 0.98) and that the best method in AlpacaFarm leads to substantial gains with human feedback. Finally, we find that AlpacaFarm can replicate qualitative behaviors of human feedback such as over-optimization of the reward model, suggesting that AlpacaFarm serves as an effective way for researchers to rapidly study and develop methods that learn from human feedback.

# 2 Background & problem statement

To begin, we introduce the instruction following task and the pairwise-comparison-based human feedback setting that we study. With this background, we formally define the goals of developing a low-cost simulator for studying instruction following and learning from human feedback.

## 2.1 Learning to follow instructions

In the instruction following task [49, 6, 76, 40], we are presented with user instructions $x \in \mathcal{X}$ (e.g. "Tell me something about Alpacas"), and our goal is to develop a model $p_\theta$ that generates high-quality responses $y \sim p_\theta(y \mid x)$ as judged by an unobserved human reward function $R : \mathcal{X} \times \mathcal{Y} \to \mathbb{R}$.

While there are a rich set of methods that learn from human feedback to directly optimize $R$ (see Appendix A), in this work we focus on the setting of *learning from pairwise feedback* (LPF) due to its central role in recent instruction-following LLMs [49]. The starting point of this process is a model that is fine-tuned on instruction-following demonstrations $(x, y)$, which we denote as $p_\theta^{\text{SFT}}(y \mid x)$. The LPF process then involves taking pairs of samples from $p_\theta^{\text{SFT}}$, querying humans for which sample within each pair is better, and learning from this pairwise feedback. Since all methods start from the SFT base, we use $p_\theta$ for notational simplicity.

**Learning from pairwise feedback (LPF).** More formally, we define the pairwise feedback dataset as $\mathcal{D}_{\text{pairwise}} = \{(x^{(j)}, y_0^{(j)}, y_1^{(j)}, z^{(j)})\}_j$. In this notation, a human annotator rates two candidate responses $y_0, y_1 \sim p_\theta(y \mid x)$ for the instruction $x$. These binary ratings $z \in \{0, 1\}$ are assumed to be generated according to their unobserved reward $R$ and $z$ indicates a (potentially stochastic) comparison for the better response $y_z$, where $R(x, y_z) > R(x, y_{1-z})$.

Many algorithms have been proposed to learn on $\mathcal{D}_{\text{pairwise}}$. Some algorithms like RLHF [18, 49] learn a surrogate reward function as the learning signal and some operate more directly on $\mathcal{D}_{\text{pairwise}}$. We defer the discussion of different learning algorithms to Section 3.4.

**Pairwise evaluation.** Once instruction-following models are trained, researchers need to evaluate these models. One common approach for evaluating models is pairwise model evaluation [15, 22, 7], which performs pairwise comparisons on outputs generated by the model $p_\theta$ and a reference model $p_{\text{ref}}$. Concretely, we collect pairwise preference for two models ($\{(x^{(j)}, y_\theta^{(j)}, y_{\text{ref}}^{(j)}, z^{(j)})\}_j$), which is aggregated by computing the average win-rate – the percentage of times $p_\theta$ is preferred to $p_{\text{ref}}$. Researchers can then compare LPF models by their win-rates against the same reference model.

## 2.2 Problem statement

The goal of AlpacaFarm is to provide three key components that enable rapid research and development of instruction following models: low-cost pairwise feedback generators, automated evaluations for methods development, and reference implementations for comparison and modification. With these three components, researchers can develop new methods in simulation and transfer these insights to build high-performance systems on actual human feedback.

For pairwise feedback, we substitute human preference judgements $z_{\text{human}} \sim p_{\text{human}}(z \mid x, y_0, y_1)$ with a simulated preference $z_{\text{sim}} \sim p_{\text{sim}}(z \mid x, y_0, y_1)$ using API LLMs. Our goal is to construct a $p_{\text{sim}}$ that is both low-cost and faithfully captures different aspects of human preference feedback, such as quality judgments, inter-annotator agreement rates, and stylistic preferences.

For evaluations, we evaluate system outputs using the pairwise preference simulator and identify evaluation datasets that reflect natural human-LLM interactions. The goal for our evaluations is to ensure that system rankings on the new evaluation dataset closely match both human rankings and rankings on instructions from real usage of the Alpaca Demo.

For reference methods, we develop and evaluate six LPF methods. Our goal will be to provide simple and working implementations that provide substantial improvements on both simulated and human feedback data. This will allow researchers to build upon and compare to competitive baselines in a complex instruction-following environment.

AlpacaFarm combines these three components into a simulation framework for learning from pairwise feedback. We evaluate the complete system by an end-to-end workflow of developing methods in simulation and transferring the insights to the real world.

Concretely, we will run each method $M$ on the simulated preferences (called $M_{sim}$) and evaluate with simulated rankings $p_{sim}$. In parallel, we will run $M$ on human preferences (called $M_{human}$) and evaluate with human rankings $p_{human}$. We consider AlpacaFarm to be successful if the simulated method rankings correlate well with the human method rankings. The rest of this work will present the details of the pairwise feedback and evaluation design (Section 3), evaluate these designs (Section 4), and analyze the different reference methods we implemented in the AlpacaFarm (Section 5).

## 3 Constructing the AlpacaFarm

In this section, we detail how we construct the AlpacaFarm. In Section 4, we then validate our design choices by comparing the LPF workflow with human feedback and evaluation.

### 3.1 Instruction following data

Before defining the details of how we simulate pairwise feedback, we must first specify a large and diverse set of instructions $x$ upon which we can build the rest of AlpacaFarm. We opt to use the Alpaca data [69] as a starting point due to its large size (52k $(x, y)$ examples) and the non-trivial instruction following capabilities of models trained on this data.

We repurpose the Alpaca data into splits suitable for learning from human feedback by following a similar splitting ratio as [49]. We created four splits (42k in total), leaving 10k for the future:

- Supervised finetuning (SFT) split: 10k data for fine-tuning the base instruction-following LLM used in subsequent steps.
- Pairwise preference (PREF) split: 10k instructions on which we collect pairwise feedback data.
- Unlabeled split: 20k unlabeled instructions used in algorithms such as PPO.
- Validation split: 2k data for development and tuning.

### 3.2 Designing simulated pairwise preference $p_{sim}$

We now describe the design of our simulated annotator for pairwise preferences. Our core proposal is to design the simulator $p_{sim}(z \mid x, y_0, y_1)$ by prompting OpenAI API LLMs. While using LLMs as a proxy for annotators has become increasingly popular [15, 39], using LLMs as part of a simulation environment poses major additional challenges. Our simulated preferences $p_{sim}$ must not only have a high agreement with human preferences $p_{human}$, it must also capture other qualitative aspects of human feedback such as inter- and intra-annotator inconsistencies. Intuitively, the noise and variability in pairwise feedback are key parts of the challenge in the LPF problem and we find that ignoring these factors leads to a simulator that diverges substantially from real-world behavior (Section 4.3).

**Basic GPT-4 prompt design.** To start with, we design prompts by providing an annotation guideline, feeding in-context examples, and leveraging batch generation to save on costs. As a first baseline, we query GPT-4 with a single prompt (denoted as $p_{sim}^{GPT-4}$) and we find $p_{sim}^{GPT-4}$ has a high agreement rate with human annotators (65%; see results in Section 4.3). However, we find that this simple baseline of $p_{sim}^{GPT-4}$ fails to capture the variability in human annotation and can lead to qualitatively different results for method development, especially for reward over-optimization (Section 4.3).

**Simulating human variability.** To more completely emulate human annotators, we modify the basic simulated annotator design to capture annotator variability in two ways. First, we emulate inter-annotator variability in the simulated pairwise preference $p_{sim}$ by mimicking a pool of annotators. We design different annotators by querying different API LLMs and varying the prompts with different formats, batch sizes, and in-context examples. In the end, we created 13 simulated annotators which we describe fully in Appendix D. Second, we emulate intra-annotator variability by directly injecting random noise and flipping the simulated preference 25% of the time.

With these ingredients, we come up with a simulated preference $p_{sim}^{ann}$ that meets our requirement of agreement and variability. Overall, annotating 1000 outputs using simulated preference only costs \$6,

which is 50x cheaper than human annotation. In Section 4, we collect actual human preference and quantitatively verify the agreement and variability of our simulated preference.

## 3.3 Designing an automatic evaluation

For researchers to develop LPF methods in the AlpacaFarm, we want to support them with an automatic evaluation so they can quickly iterate while reliably comparing methods. To replace the usual human evaluation, there are two challenges. First, how do we quantify the quality of different models? Second, what instructions can we use that are representative of human interactions?

**Evaluation protocol.** To quantify the quality of an LLM $p_\theta$, we measure the win-rate of that LLM against a reference model, i.e, the expected number of times that the output from $p_\theta$ is preferred to the output of a reference model $p_{ref}$ on the same instruction $x$. The benefits of using simulated win-rates are that it provides a metric that is easy to understand, is comparable across methods, and can reuse the routine we built for pairwise feedback. We use the 13 simulated annotators described in Section 3.2 without injecting the additional noise (as adding uniform noise does not change model rankings) and denote this preference simulator as $p_{sim}^{eval}$. For the reference model, we use Davinci003 as it is a well-studied system that performs similarly to the models we fine-tune.

**Evaluation data.** Instruction following requires diverse coverage over realistic interactions. To build an appropriate evaluation protocol, we combine several open-source evaluation datasets and use real-world interactions with a demo instruction-following LM (Alpaca Demo [69]) as guidance for constructing our data combination. Due to privacy concerns, we do not directly release the demo data and opt to use it to guide how we combine existing open evaluation datasets.

Our final evaluation set consists of 805 instructions, which includes 252 instructions from the self-instruct test set [75], 188 from the Open Assistant (OASST) evtest setaluation, 129 from Anthropics' helpful test set [6], 80 from Vicuna test set [80, 17], and 156 from Koala test set [22]. In **??**, we show example instructions from our evaluation set (Table 7) and their root verb distribution(Figure 13), which shows the diverse coverage of our data. We find that aggregating across datasets is important for automatic evaluation to match real-world interactions, as discussed in Section 4.4.

## 3.4 Reference methods in AlpacaFarm

Finally, AlpacaFarm defines a collection of validated LPF methods for instruction following. We describe those methods in detail in Appendix B but provide a brief overview here. In all LPF methods that follow, we begin by first performing an initial fine-tuning step on supervised data of instructions and outputs.We begin by describing two simple baselines that directly operate on pairwise feedback.

- **Binary FeedME.** Binary FeedME [47] continues supervised fine-tuning on the preferred output in each pairwise comparison.
- **Binary reward conditioning.** Reward conditioning [28, 36, 30] is a simple scheme that incorporates learning from negative examples; a token denoting whether the output was preferred is prepended before fine-tuning, and the positive token is used to condition at inference time.

Many LPF methods do not directly operate on pairwise feedback data, but instead first construct a surrogate reward model by fine-tuning a classifier from the SFT base using pairwise feedback. The following LPF methods maximize the continuous-valued reward defined by the logits of this classifier.

- **Best-of-$n$ sampling.** Best-of-$n$ (or re-ranking) [67, 5, 23, 9] is a simple but effective inference-time method that draws $n$ i.i.d. responses from the SFT model and returns the response with the highest surrogate reward. Unless stated otherwise we use $n = 1024$ in our experiments.
- **Expert iteration.** Expert iteration [2, 64, 74] is the natural training-time extension of best-of-$n$: it first generates according to best-of-$n$ on new instructions and then fine-tunes on the best outputs.
- **Proximal Policy Optimization (PPO).** PPO [26, 62] is a reinforcement learning algorithm that maximizes surrogate reward, subject to a KL penalty keeping parameters near SFT initialization.
- **Quark.** We use the top-quantile variant of Quark [42] which bins sequences by reward and trains on the best bin, along with adding KL and entropy regularization.

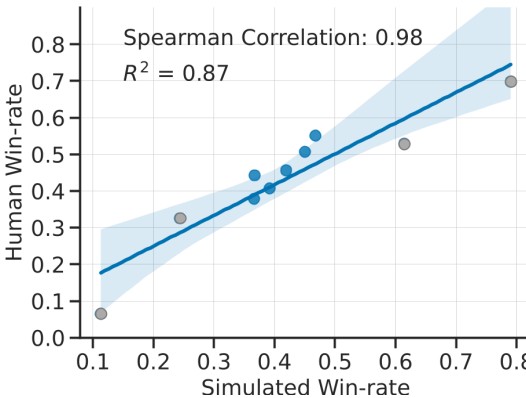

Figure 2: The ranking of methods trained and evaluated in AlpacaFarm matches that of methods trained and evaluated in the human-based pipeline. Each blue point represents one method $M$ (e.g. PPO). The x-axis shows the simulated evaluation (win-rates measured by $p_{\text{sim}}^{\text{eval}}$) on methods trained in simulation $M_{\text{sim}}$. The y-axis shows human evaluation (win-rates measured by $p_{\text{human}}$) on methods trained with human feedback $M_{\text{human}}$. Gray points show models that we did not train, so their $x$ and $y$ values only differ in the evaluation (simulated vs human). Without those points, we have $R^2 = 0.83$ and a Spearman Correlation of $0.94$.

## 4 Validating the AlpacaFarm simulator

With the simulator and methods defined, we now evaluate AlpacaFarm. As our main result, in Section 4.2 we analyze the correlation between the final rankings of methods in both the simulated LPF workflow and human-based LPF. Afterward, we will analyze more specifically whether our pairwise feedback accurately mimics human pairwise feedback (Section 4.3) and whether rankings on our evaluation data match rankings on the Alpaca Demo data (Section 4.4).

### 4.1 Experimental details

**Models.** As baseline and starting point for LPF methods, we fine-tuned LLaMA 7B on the 10k SFT split. We take SFT 10k as starting point for all LPF methods and collect the simulated preference $p_{\text{sim}}^{\text{ann}}$ and human preference $p_{\text{human}}$ from SFT 10k's outputs (with `temp=1.0`) on the 10k instruction PREF split. Then, for each of the six reference LPF methods $M$:

- We trained and tuned $M$ on simulated preferences $p_{\text{sim}}^{\text{ann}}$, evaluating the resulting model $M_{\text{sim}}$ against the Davinci003 reference with the simulated evaluator $p_{\text{sim}}^{\text{eval}}$.
- We trained a few models $M$ on human preferences across hyperparameter ranges identified in simulation, evaluating the resulting model $M_{\text{human}}$ against Davinci003 with humans $p_{\text{human}}$.

In addition to the six methods, we also evaluate existing instruction-following and base models: GPT-4 (gpt-4-0314), ChatGPT (gpt-3.5-turbo-0301), Davinci001 (text-davinci-001), LLaMA 7B [72], and Alpaca 7B [69]. Alpaca 7B is a LLaMA 7B model finetuned on all data splits, denoted SFT 52k. For these models, we measure both the simulated win-rate $p_{\text{sim}}^{\text{eval}}$ and human win-rate $p_{\text{human}}$.

At inference time, for all systems except best-of-$n$, we sample with `temp=0.7` and set the maximum number of tokens to be 300. For best-of-$n$ sampling, we found a higher temperature to be helpful in encouraging output diversity, and so we rerank samples from SFT 10k with `temp=1.0`. We provide more thorough experimental details and hyperparameters for all methods in Appendix C.

**Human annotation.** We collected reference human annotation by showing crowd-workers two potential outputs $y_0$ or $y_1$ for a given instruction $x$ and asked them to select the index $z \in \{0, 1\}$ of their preferred output. Annotators are recruited from Amazon Mechanical Turk using a qualification test of 25 questions. Out of an initial pool of 34 annotators, we selected the 16 whose agreement rate was higher than 70% with the author's annotations. We paid the annotators a median hourly rate of $21, leading to a one-time $3000 cost of annotating our PREF split and a recurring $242 cost for evaluating a single model on the 805 evaluation instructions. See Appendix E for additional details.

### 4.2 End-to-end validation of AlpacaFarm

We now analyze the correlation between rankings in simulation and on human data. Figure 2 shows the win-rate of methods in AlpacaFarm (x-axis) with the win-rate from the human-based pipeline (y-axis). We see that the rankings have a Spearman correlation of 0.98, which suggests that AlpacaFarm faithfully captures the rankings among different LPF methods. This enables researchers to develop models at low-cost in AlpacaFarm and transfer insights to train models on real-world human interactions.

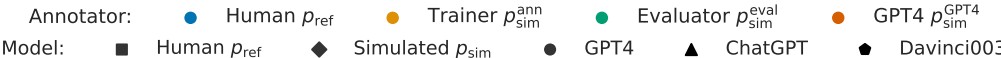

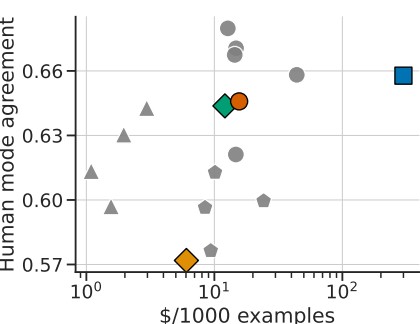

Figure 3: Our simulated annotators are cheap and have a high agreement with human annotators. We show price (x-axis) vs agreement (y-axis) of each annotator with the majority vote among 3 human annotations. Grey points are the simulated annotators in the pool, the green ◆ shows the resulting pool of annotators (used for evaluation), the orange ◆ shows the same pool with random noise added during training. This does not change the implied reward function from ◆, but makes the learning problem more challenging. The blue ■ shows the average of human annotators, and the red ● shows a single low variance GPT-4 annotator analyzed below.

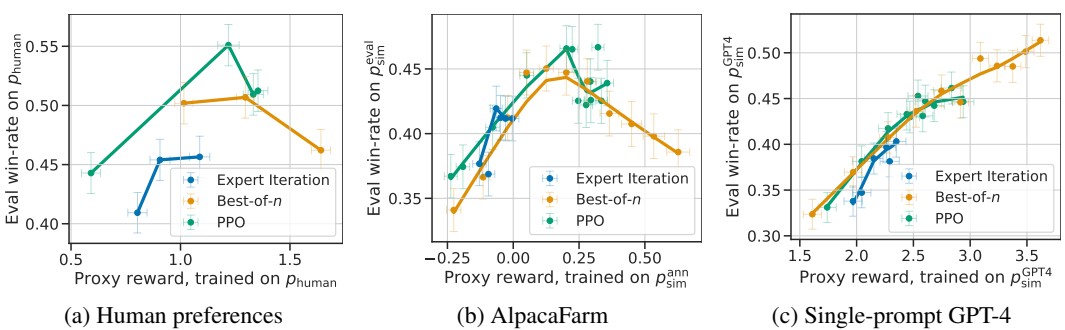

(a) Human preferences        (b) AlpacaFarm        (c) Single-prompt GPT-4

Figure 4: Human and AlpacaFarm preferences result in over-optimization, while a naive single-prompt GPT-4 simulator does not. Left: training and evaluation with human preferences ■. Middle: training and evaluation with AlpacaFarm preferences ◆, ◆. Right: training and evaluation with single-prompt GPT-4 preferences ●. The x-axis measures the average surrogate rewards on the eval set.

Inspecting these results more closely, we point out the two rank mismatches. The first comparison is SFT10k against SFT52k, where human annotators preferred SFT10k (44.3% vs 40.7%) while the simulator had the opposite preference (36.7% vs 39.2%, Table 1). The other mismatch is ChatGPT against PPO, where human annotators preferred PPO (55.1% vs 52.9%) unlike the simulator (46.8% vs 61.4%). In both cases, these are not major mistakes, as we do not expect SFT52k to be much worse than SFT10k or for a 7B LLaMA model to substantially outperform ChatGPT.

## 4.3 Validating the pairwise preferences component

Having demonstrated that AlpacaFarm succeeds at the end-to-end validation of methods rankings, we now take a closer look at our pairwise preferences, showing that they agree with human annotators and replicate important qualitative features of model training. For additional details see appendix D.

**Simulated annotators match human agreement.** We begin by computing agreement levels between our simulated annotator and a majority vote of 3 human annotators, comparing this to the agreement level of a held-out human annotator, as shown in Figure 3. We find that our evaluator $p_{sim}^{eval}$ (◆) has a 65% agreement rate with the human majority vote, which is similar to the held-out human agreement rate at 66% (■). At the same time, $p_{sim}^{eval}$ is $25\times$ cheaper ($300 \rightarrow $12 per 1000 examples). The training time annotator $p_{sim}^{ann}$ (◆) has lower agreement due to label flip noise but this does not mean that $p_{sim}^{ann}$ is less faithful to human annotations, since this noise is unbiased and both annotators ($p_{sim}^{ann}$, $p_{sim}^{eval}$) represent the same underlying preference function.

Figure 3 also shows that we identified single prompts performing even better than $p_{sim}^{eval}$, with one prompts achieving 68% agreement. While this high agreement level is impressive, we do not use single

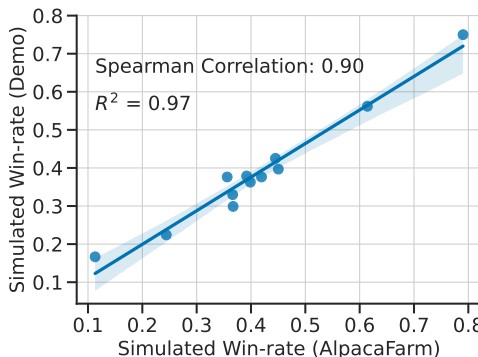

Figure 5: Correlation plot of simulated win-rates computed on AlpacaFarm's evaluation versus that on real-world interactions with the Alpaca Demo.

prompts for AlpacaFarm, as single prompts do not replicate the inter-annotator variability important for a simulator. Instead, randomizing over different annotators and injecting additional noise is needed to match the distributional features and learning dynamics of human data, which we discuss next.

**Simulated annotators replicate overoptimization.** We now show that modeling annotator variability in the simulated annotator ($p_{\text{sim}}^{\text{ann}}$) is necessary to capture important qualitative features of LPF model training. To do so, we compare the behavior of three of the best-performing models trained under $p_{\text{sim}}^{\text{ann}}$ with those trained using the single GPT-4 prompt $p_{\text{sim}}^{\text{GPT-4}}$, which has higher human agreement but little annotator variability.

Figure 4 shows the learning dynamics of these models for pairwise feedback by $p_{\text{human}}$ (left), $p_{\text{sim}}^{\text{ann}}$ (middle), and $p_{\text{sim}}^{\text{GPT-4}}$ (right) as the PPO iteration count and rerank sample counts for best-of-$n$ and expert iteration are increased. For both the human and AlpacaFarm preferences, models that more effectively optimize the surrogate reward (x-axis) improve in win-rate (y-axis) up until a point, where overoptimization of the reward occurs and win-rates decrease. In contrast, simple GPT-4 feedback shows no overoptimization, leading to the false conclusion that the LPF methods can be optimized for much longer than is actually the case. For example, Figure 4 right shows best-of-1024 to be much better than PPO, which disagrees strongly with the results on human data.

Noted in prior work [21], overoptimization is the result of the proxy reward model $\hat{R}_\phi$ being an imperfect estimate of the true reward $R$. We hypothesize that the strong overoptimization seen for human annotators is partly due to the (inter- and intra-) variability of their annotations, which degrades the quality of the reward model. To test this hypothesis, we measured the *variance* of each of the three annotators by calculating the average error of a held-out annotation to the majority vote over 3 random draws. At 0.26 for $p_{\text{sim}}^{\text{eval}}$ and 0.43 for $p_{\text{sim}}^{\text{ann}}$, AlpacaFarm is close to the human variance of 0.35; in contrast, the GPT-4 annotator has a much lower variance at 0.1. Finally, in Appendix F.1 we ablate the AlpacaFarm design more finely and find that the added label noise provides the majority of the benefit for inducing overoptimization. Appendix D contains further analyses of bias and variability of annotators.

### 4.4 Validating the evaluation protocol

Finally, we test our evaluation data that combines existing open-source evaluation datasets. While we have observed that this dataset is diverse (see Figure 13), it is unclear whether it evaluates performance for any type of real-world human usage. To resolve this, we measure method-level correlations against a set of real user interactions recorded on the Alpaca Demo [69]. We manually went through the interactions and identified 200 instructions that do not contain any personal identifying information, toxic or unsafe questions, and those that refer to the chatbot directly (e.g. "who are you developed by?"). The terms of use for the demo do not allow us to publicly release this data, but we use this data to evaluate the proposed evaluation set.

We use the 11 systems displayed in Figure 2, with the LPF methods trained in simulation, and evaluate them using $p_{\text{sim}}^{\text{eval}}$. Figure 5 plots the simulated win-rates on the Demo instructions against those on the AlpacaFarm evaluation data. The two win-rates are strongly correlated ($r^2 = 0.97$), indicating that AlpacaFarm evaluation data can serve as a proxy for evaluating methods on simple demo interactions.

Table 1: AlpacaFarm evaluation results on baseline and LHF methods. For methods without a *, the left column shows win-rates when we train and evaluate in simulation, while the right column shows win-rates when we train and evaluate with human feedback. For models with a *, those are not trained by us so the left and right columns respectively show simulated and human evaluation. Win-rates are computed against Davinci003. We did not evaluate Quark and Binary Reward Conditioning for human evaluation because they underperformed in development. † ChatGPT and GPT-4 were prompted for responses shorter than 1000 characters to have length comparable to other methods.

| Method | Simulated Win-rate (%) | Human Win-rate (%) |
|---|---|---|
| GPT-4*† | $79.0 \pm 1.4$ | $69.8 \pm 1.6$ |
| ChatGPT*† | $61.4 \pm 1.7$ | $52.9 \pm 1.7$ |
| PPO | $46.8 \pm 1.8$ | $55.1 \pm 1.7$ |
| Best-of-1024 | $45.0 \pm 1.7$ | $50.7 \pm 1.8$ |
| Expert Iteration | $41.9 \pm 1.7$ | $45.7 \pm 1.7$ |
| SFT 52k | $39.2 \pm 1.7$ | $40.7 \pm 1.7$ |
| SFT 10k | $36.7 \pm 1.7$ | $44.3 \pm 1.7$ |
| Binary FeedME | $36.6 \pm 1.7$ | $37.9 \pm 1.7$ |
| Quark | $35.6 \pm 1.7$ | - |
| Binary Reward Conditioning | $32.4 \pm 1.6$ | - |
| Davinci001* | $24.4 \pm 1.5$ | $32.5 \pm 1.6$ |
| LLaMA 7B* | $11.3 \pm 1.1$ | $6.5 \pm 0.9$ |

## 5 Benchmarking reference methods on the AlpacaFarm

We now study the performance of reference methods in AlpacaFarm. Table 1 contains the details of the main evaluation results (presented in Figure 2). In the rest of this section, we discuss our findings from these results, demonstrating that the conclusions we reach using human feedback could have been derived using AlpacaFarm at a substantially lower cost. See Appendix C for experimental details.

### 5.1 Comparing LPF methods

**Supervised fine-tuning is highly effective.** Table 1 shows that the SFT is effective and provides the majority of gains. SFT brings the base LLaMA model up from a simulator win-rate of $11\%$ to $37\%$ and a human win-rate of $7\%$ to $44\%$. However, we observe no gain from SFT 10k to SFT 52k.

**PPO tops the LPF leaderboard.** Among the LPF methods we study, PPO performs the best in both the simulator ($47\%$) and on human data ($55\%$). Notably, with a win-rate of $55\%$, the PPO trained with human feedback was preferred to ChatGPT by our annotators for single-turn instruction-following. This surprising result is likely because ChatGPT was prompted for responses shorter than 1000 characters and both crowd-workers and simulators annotators preferred longer responses (see appendix D.2). For example, in AlapcaEval [35] (slightly different simulated annotator) we show that the unconstrained ChatGPT achieves $89\%$ win-rate, much higher than the $44\%$ win-rate for the PPO model.

**Best-of-$n$ is simple and competitive.** Best-of-$n$ sampling outperforms all LPF methods besides PPO. The performance best-of-$n$ sampling points to the useful learning signals captured by the reward model and helps us understand how training-time LPF methods benefit from a reward model.

**Expert Iteration and Quark lag behind.** Even though expert iteration trains with best-of-$n$ outputs, it trails best-of-$n$ with a 3-6% win-rate gap in both simulated and human feedback workflows. This result suggests that acquiring the improvement from best-of-$n$ sampling is not straightforward and points to why more complex learning algorithms like PPO might be useful. For Quark, although the rewards of samples improve during training, the resulting models did not outperform the SFT baseline.

**Methods that directly learn from pairwise feedback do not perform well.** We observe that binary reward conditioning and binary FeedME do not improve over the SFT 10k baseline. This result suggests that learning a surrogate reward model may be an important ingredient to LPF.

In appendix F.3 we analyze the outputs of different models. We find a major difference is that LPF methods produce longer outputs in both human and simulated feedback.

## 5.2 Using AlpacaFarm to train models directly for human deployment

A natural question is whether models trained in AlpacaFarm can directly perform well on human evaluation Table 2 shows that AlpacaFarm can be repurposed for this goal by using simulated annotators that maximize agreement rather than match human variability, i.e., using preferences from the single low-variance GPT-4 annotator $p_{\text{sim}}^{\text{GPT-4}}$ rather than our noisy annotators $p_{\text{sim}}^{\text{ann}}$.

Table 2: Model transfer results.

| Method | Human Win-rate (%) |
|---|---|
| $\text{PPO}_{\text{human}}$ | 55% |
| $\text{Best-of-16}_{\text{human}}$ | 51% |
| $\text{PPO}_{\text{sim}}^{\text{GPT-4}}$ | 50% |
| SFT 10k | 44% |
| $\text{PPO}_{\text{sim}}^{\text{ann}}$ | 43% |

Table 2 compares the best model $\text{PPO}_{\text{sim}}^{\text{ann}}$ trained on $p_{\text{sim}}^{\text{ann}}$ with a similar model $\text{PPO}_{\text{sim}}^{\text{GPT-4}}$ trained on the GPT-4 annotator. We find that $\text{PPO}_{\text{sim}}^{\text{ann}}$ only achieves 43% win-rate, while $\text{PPO}_{\text{sim}}^{\text{GPT-4}}$ achieves 50% win-rate. As context, we show win-rates of the initial SFT, the human PPO $\text{PPO}_{\text{human}}$, and the best non-PPO human method $\text{Best-of-16}_{\text{human}}$. Overall, training in AlpacaFarm can provide good models for deployment, but still suffers a 5% performance gap relative to real human annotations.

## 6 Limitations

**Validation.** Section 4 validates the use of AlpacaFarm, but there are nevertheless some limitations with the validation setting. First, the instructions we considered (even those from the real world-demo in section 4.4) are relatively simple and single-turn. Second, all models we fine-tuned use a LLaMA 7B as starting point. Finally, Human validation is based on feedback from 13 crowd-workers, which may not reflect broader human preferences and seem to have biases such as preferences for longer outputs as highlighted in appendix D.2. Although we do not expect the previous limitations to significantly affect the usefulness of AlpacaFarm, we encourage users to be vigilant when using any simulator and hope to further validate AlpacaFarm as more datasets and base models become available.

**Assumption of an oracle LLM.** AlpacaFarm assumes that we can access an oracle LLM, much more powerful than the investigated models, that can be used to approximate human feedback. While this may be true in research settings, it is not always the case in practice. We believe that investigating the use of the same model for the simulator and for learning is a promising direction for future work.

**AlpacaFarm for fine-grained development.** Section 4 shows that AlpacaFarm is effective for model selection and can replicate important behaviors such as over-optimization. We have nevertheless found that suitable hyperparameters for learning algorithms can be different for training with simulated feedback compared to human feedback. For example, due to changes in the scale of values of the surrogate reward model, the range of suitable KL regularization coefficients for RLHF is different.

**Simulated annotators.** A key component of AlpacaFarm is the simulated annotators. In section 4.3 and appendix D we showed that our annotators have high agreement with human annotators and exhibit similar biases, such as preferences for longer outputs and lists. There are nevertheless some biases that are specific to simulated annotators. For example, we found that simulated annotators prefer the first outputs shown in the prompt and outputs that come from the same model as the simulator. Although we controlled for those biases by randomizing the order of output in the prompt, and using the same training data for all considered methods, there may be other biases that we have not identified. For better automatic annotators and further analysis refer to AlpacaEval [35].

## 7 Related works and outlook

The idea of building a simulator is closely related to recent efforts in automatic model evaluations [16, 55, 15, 53, 37] and constitutional AI [8]. However, our work crucially differs in our goal of building a simulator, which requires us to mimic distributional properties such as inter-annotator disagreement. We discuss these differences and the broader context of our work in Appendix A.

We showed that AlpacaFarm substantially lowers the cost and iteration time for developing methods that learn with pairwise feedback. AlpacaFarm provides a blueprint for constructing simulators for AI research that requires human supervision, and we view it as an exciting opportunity to expand this simulation approach to data from other domains and alternative forms of human feedback.

## Acknowledgments and Disclosure of Funding

We thank the Stanford Center for Research on Foundation Models (CRFM), Stanford HAI, and Stability AI for compute support. Data collection efforts were supported through the Tianqiao & Chrissy Chen Institute and an Open Philanthropy grant. XL is supported by a Stanford Graduate Fellowship. RT is supported by the NSF GRFP under Grant No. DGE 1656518. YD is supported by a Knights-Hennessy Scholarship.

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

# A Related work

**Instruction following.**    There have been an number of works studying instruction following as a cross-task generalization across a pool of NLP tasks [44, 77, 58, 4, 76]. Our work focuses on a recent trend in instruction following methods which increasingly focus on real world human interaction patterns [49, 6], rather than collections of existing NLP benchmarks. For example, InstructGPT was developed on user instructions submitted to OpenAI API [49]. Our work builds upon these works by attempting to bridge the gap between the ease of development and evaluation of traditional academic benchmarks and the more complex algorithms and real-world settings of recent works on instruction following.

**Simulating human feedback.**    Constitutional AI [7] simulates human feedback with AI feedback for model development to improve harmlessness and helpfulness. AlpacaFarm, on the other hand, simulates human feedback with API LLMs so that simulated experiments reflect the outcomes of experiments performed with real human feedback. Due to the difference in goals, the construction and usage of the feedback simulator are different in the two settings. For example, AlpacaFarm's simulator perturbs LLM preferences with bit-flip label noise to mimic the noisiness of human annotation, whereas Constitutional AI's simulator doesn't inject extra noise.

The evaluation aspects of our work are related to a growing line of work on simulating human annotation for evaluation [15, 56, 15, 54, 38, 39]. Our core evaluation and feedback mechanism makes use of the same underlying ideas, but our work is distinguished by a focus on using pairwise feedback for training, as well as careful validation beyond per-example agreement metrics. AlpacaFarm shows that LLM feedback can capture method-level correlations as well as qualitatively important features of human annotation for phenomena such as overoptimization.

Our goal of emulating human annotators also connects to work on simulating humans with LMs based on personas [51, 50, 1, 3], as well as works that simulate human behavior in the context of cognitive science, social science, and economics [73, 27]. Our work complements these works by showing that simulated LLM annotators can replicate many of the qualitative features of training on pairwise human feedback.

More broadly, building a simulator environment to enable low-cost experimentation is common in the field of reinforcement learning and robotics [12, 71, 70, 68, 19, 25, 20]. Our work shares the same underlying motivations, but instead of simulating physical systems, AlpacaFarm simulates human preference feedback.

**Methods for learning from feedback.**    To hold annotation cost constant across learning methods, we have focused only on methods that learn from pairwise feedback in this work. However, there exist methods in the literature other than those explored in AlpacaFarm that can incorporate alternative sources of feedback such as natural language [78, 34, 24, 63, 59, 14, 60, 43], numeric ratings [47, 33], or execution traces [14]. We view extensions of AlpacaFarm to these settings as exciting future work.

We have included a set of RL algorithms in our study that optimize the surrogate reward, but this set is by no means comprehensive. RL research applied to NLP has a long history [79, 66, 29, 52, 45, 32, 31, 57, 65], and we expect future work in this direction to benefit from the ideas and artifacts in AlpacaFarm.

# B Reference LPF methods on AlpacaFarm

We provide a more thorough description of the methods in AlpacaFarm along with the custom modifications we make. For more on hyper-parameter tuning, experiment details, and further ablations, please see Appendix C.

Our methods fall into two categories based on whether they fit a surrogate reward model as part of the learning process. In addition, to those methods which are trained in AlpacaFarm, we also include well-known baseline methods from the OpenAI API.

**API baseline methods.** In both the human and the simulated feedback workflows, we evaluate the following methods from the OpenAI API: GPT-4, ChatGPT, Davinci001, and implicitly Davinci003, which is the baseline model we compare every model with. Outputs from all these models are sampled at temperature 0.7 and with top-$p$ 1.0. For the text models, we use the same prompt as all other reference methods in AlpacaFarm. For both chat models (ChatGPT and Davinci003), we had to change prompts for two reasons. First, those models require prompts in a chat format, which is different from the text format. Second, we found that those models generated sequences that were much longer than the rest of our models, which were trained to output sequences of less than 300 tokens. We thus ask in the system prompt for a response that is shorter than 1000 and 500 characters respectively for ChatGPT and GPT-4, which we found to give shorter answers while working similarly to a raw prompt that does not mention the length.

## B.1 Methods that directly learn from pairwise feedback

**Binary FeedME.** FeedME is a method proposed by OpenAI [47] that incorporates human feedback with supervised fine-tuning on model generations that are rated 7/7 by human labelers. We adapt this approach to the pairwise feedback setting and call this baseline binary FeedME. This approach fine-tunes the SFT model on the chosen response in each preference pair with supervised learning.

**Binary Reward Conditioning.** Motivated by controllable generation through conditioning [28, 36, 30, 22], we propose binary reward conditioning, a baseline method that fine-tunes the SFT model with the feedback data $\mathcal{D}_{\text{pairwise}}$ by conditioning instances with either a positive or negative control token. Specifically, for each instance $(x, y_0, y_1, z) \in \mathcal{D}_{\text{pairwise}}$, the string concatenation of instruction $x$ and response $y_z$ denoted as $[x, y_z]$ is prepended with the positive token and used in supervised fine-tuning (similarly $[x, y_{1-z}]$ is prepended with the negative token). This process creates a modified demonstration dataset that is double the size of $\mathcal{D}_{\text{pairwise}}$. At test time, we draw samples from the fine-tuned model conditioned on the positive token.

## B.2 Methods that optimize a surrogate reward function

We now describe methods that incorporate feedback by first building a surrogate reward model with pairwise feedback data. To start, we describe the step of training the surrogate reward model.

To train a parameterized surrogate $\hat{R}_\phi$, one can maximize the log-likelihood of the preferences $z$ under the Bradley-Terry model [11]

$$\text{maximize}_\phi \sum_j \log P(z^{(j)} \mid x^{(j)}, y_0^{(j)}, y_1^{(j)}) = \sum_j \log \frac{\exp(\hat{R}_\phi(x^{(j)}, y_z^{(j)}))}{\exp(\hat{R}_\phi(x^{(j)}, y_0^{(j)})) + \exp(\hat{R}_\phi(x^{(j)}, y_1^{(j)}))}. \tag{1}$$

Once the surrogate reward model is trained, both training and inference algorithms can optimize against the reward model rather than query pairwise feedback. While this can be a powerful approach, we will see that it can also lead to *over-optimization* [21] where models learn to exploit the reward model rather than achieve high true reward. We now describe 4 methods that leverage the surrogate reward model.

**Best-of-$n$ Sampling.** Best-of-$n$ sampling (or re-ranking) [67, 5, 23, 9] is a common inference-time method that aims to improve the generation quality. Given an input $x$, the method returns the response with the highest surrogate reward value among $n$ i.i.d. responses drawn from the SFT model. While simple to implement and useful as a baseline, this approach incurs high inference costs.

**Expert Iteration.**    Expert iteration [2, 64, 74] is a technique that has recently been used to train language models. We adapt this approach in AlpacaFarm as a two-step method. In the first step, we perform best-of-$n$ sampling and store the generated samples. In the second step, we fine-tune $p_{\text{SFT}}$ on these samples with supervised learning. While prior work applies expert iteration for multiple rounds by performing best-of-$n$ sampling again for intermediate models, we focus on performing a single round. In Appendix C, we include our preliminary study of multi-round expert iteration.

**Proximal Policy Optimization.**    Proximal Policy Optimization [PPO; 26, 62] is a popular RL algorithm that has been recently used to develop InstructGPT [49] and ChatGPT [46]. When applied to fine-tune LMs with RLHF, PPO maximizes the following KL-regularized objective w.r.t. model parameters $\theta$

$$\mathbb{E}_{x \sim p(x),\ y \sim p_\theta(y|x)} \left[ \hat{R}_\phi(x, y) - \beta \log \frac{p_\theta(y \mid x)}{p_{\text{SFT}}(y \mid x)} \right], \tag{2}$$

where $p(x)$ is an unlabeled instruction distribution, $p_\theta(y \mid x)$ is fine-tuned from the $p_{\text{SFT}}$ model, and $\beta \in \mathbb{R}$ is a regularization coefficient. Each step of PPO alternates between drawing samples from the current policy and performing gradient updates based on the pool of samples with importance sampling and clipping.

**Quark.**    Quark is inspired by reward conditioning and has been shown to be effective for controllable generation tasks. Like binary reward conditioning, Quark on train sequences with prepended control tokens. Unlike binary reward conditioning, Quark bins model samples into multiple groups based on the reward value, adds KL and entropy regularization, and repeats the entire process across multiple rounds.

In our preliminary analysis, we find the top-quantile variant reported in [41], i.e., only training on the best reward group, to perform the better than the all-quantiles variant which trains on all groups.

# C    Details on methods implementation and hyperparameters

For all hyperparameters and code see `https://github.com/tatsu-lab/alpaca_farm`

## C.1    PPO

We follow an existing PPO implementation for fine-tuning language models,[2] but also introduce modifications. First, off-the-shelf PPO implementations for language model fine-tuning tend to normalize the estimated advantage for each minibatch. We found this led to training instabilities for small minibatch sizes and instead normalize the advantage across the entire batch of rollouts obtained for each PPO step. Second, we initialize the value model from the reward model as opposed to the SFT model, following more recent documented practice [49] (the authors did not release code). Our preliminary experiments showed that initializing from reward worked much better than initializing from SFT for maximizing the surrogate reward.

We tuned hyperparameters to improve training stability and reduce convergence time so that experiments can reliably finish with relatively tight compute budgets. In the end, we settled on a batch size of 512 for each PPO step, which consisted of 2 epochs of gradient steps each performed with a batch of 256 rollouts. We used a peak learning rate of $10^{-5}$ which decayed to 0 throughout training. We clipped the gradient by Euclidean norm with a threshold of 1. We trained for 10 full passes over the unlabeled set, which amounts to 390 PPO steps. Performance typically peaked very early on during training (see Figure 4). We set $\lambda$ and $\gamma$ both to 1 for generalized advantage estimation [61]. We used a fixed KL regularizer coefficient as opposed to an adaptive one. We tuned the coefficient value for both simulated and human PPO, and settled with 0.02 for human PPO, and 0.002 for simulated PPO. We note that suitable values for the KL regularizer coefficient depend on the early stopping criteria and the scale of surrogate reward values.

## C.2    Quark

We re-implement Quark for our needs and make several modifications. First, the original Quark formulation accumulates rollouts during training and stores them in a pool that consistently grows. We found this led to overhead that increased during training (since after each rollout batch is generated, the pool is expanded and rollouts in the pool are re-sorted by their reward values). To operate under a reasonable compute budget, we discard previous rollouts once a new batch of rollouts is generated. In other words, the pool is reset once rollout is performed. This modification made the compute cost constant throughout training and thus more predictable overall. Second, we found that training on rollouts of more bins led to worse efficiency for reward optimization, and thus opted to train only on rollouts of the top-scoring bin (best-quantile variant in the original paper [42]). Preliminary ablations on a simple sentiment task showed that any potential loss in perplexity for the best-quantile variant can be compensated by turning up the KL regularizer. Lastly, we found the entropy penalty used in the original Quark formulation to give no benefit for working with instruction following. Small entropy penalty terms were enough to cause big degradations in text generation quality in terms of fluency.

For the official run with reported results, we used a KL regularizer coefficient of 0.05, a peak learning rate of $3 \times 10^{-6}$ which decayed to 0 throughout training. Each Quark step had batch size 512 for rollout, and 2 epochs of gradients updates each with batch size 256. We clipped the gradient by Euclidean norm with a threshold of 1. We trained for 10 full passes over the unlabeled set, which amounts to 390 Quark steps.

---

[2] `https://github.com/openai/lm-human-preferences`

# D  Pairwise preference simulation

## D.1  Details about simulated annotators

For all our simulated annotators we used OpenAI API to generate outputs. We first discuss below the overall design choices for all our simulators below, and then discuss our annotator pool below in more detail. For all the actual prompts we used refer to https://github.com/tatsu-lab/alpaca_farm.

**Randomized order.**    For each annotator, we randomize the ordering between the two outputs to annotate, i.e., we randomly choose which output is the first and which is the second. We found randomization to be important given that the first output is often preferred by simulated annotators.

**Prompts with and without inputs.**    Following the Alpaca dataset [69] and self-instruct framework [75] some instructions have associated inputs, while others do not. For each annotator, we thus write two corresponding prompts, one for instructions with inputs and one for instructions without inputs. Both prompts are essentially the same but in-context examples differ in the presence of the input.

**Batching for GPT4.**    When adding in-context examples, prompts can become relatively long, which leads to high-cost and waiting time when using GPT-4 as a simulator. To decrease cost and increase annotation speed, we amortize the cost of in-context examples by providing a batch of instruction-output pairs to annotate at once by GPT-4. For our simulated annotator we use a maximum batch size of 5 but found during development that we could fit batch size up to 20 in the context window without significantly decreasing performance. To improve performance when using batching, we found it useful to provide a few in-context examples in a batched format and to index every component of an annotation (instruction, input, output, . . . ).

**Improving parsing for ChatGPT.**    Overall we found ChatGPT to be much more sensitive and harder to use as a simulator. In particular, we found it to be more sensitive to the prompt format and to often fail to generate annotations that could be parsed, e.g., by responding "Neither is better, this depends on personal preferences" despite being explicitly instructed to choose a preference. We found two tricks to be effective to make ChatGPT's more parsable. First, we add a negative bias to tokens such as "Neither" and "Both" and a positive bias to the tokens that we hoped to match. We found the aforementioned biasing of tokens to work well but it can be problematic when using Chain of Thought reasoning. A second trick that we found to be effective is to ask ChatGPT to generate a JSON object that contains a string field with a short explanation (Chain of Thought) and a boolean field that indicates whether the first output was preferred.

Now that we have discussed the overall design choices for our simulated annotators, we discuss in more detail the prompts and parameters for each of our annotators.

**AlpacaFarm's evaluation annotators $p_{\text{sim}}^{\text{eval}}$.**    To try to match the bias and variance of human annotators, we use a pool of 13 simulated annotators that were developed at different stages of the project. In particular, we use the following sources of variations:

- Models. Five of the annotators are powered by GPT-4, four by ChatGPT, and four by Davinci003. The difference between different annotators for the same model is mostly the prompt.

- In-context examples. Prompts for the same models use different numbers of in-context examples.

- Prompt format. We use different prompt formats between and for the same model. For example different batch sizes and different formats of outputs (JSON vs raw text).

- Preferences. Two of the GPT4 annotators are explicitly prompted to prefer sequences that are respectively long and short.

- Sampling. For each annotator in the pool, we use a sampling temperature of 1.0 with top $p$ also 1.0. The high temperature means that we have variability that arises from sampling.

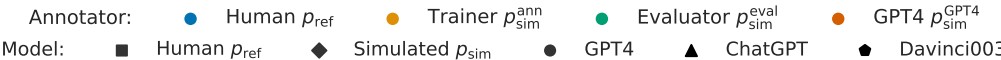

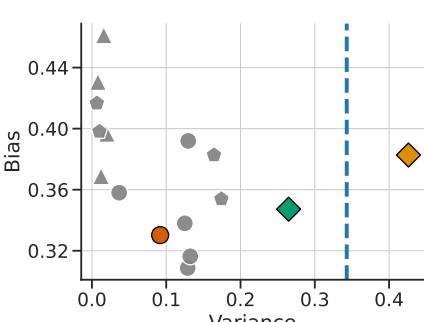

Figure 6: Our simulated annotators achieve relatively low bias with human annotators and match human variance. The y-axis shows the estimated bias, i.e., the error between the majority vote of 4 simulated annotators and the majority vote of 4 human annotators. The x-axis shows the estimated variance, i.e., the error between a held-out annotation and the majority vote of the other three annotators. The bias of humans is by definition 0, and variance is shown with a blue line. Grey points are all the annotators in our simulated pool, the green point shows the resulting pool of annotators (which we use for evaluation), the orange point shows the same simulated pool with additional noise (which we use for training), the blue point the average human annotator, and the red point shows a single low variance GPT-4 annotator we analyze.

**AlpacaFarm's training annotators $p_{\text{sim}}^{\text{ann}}$.** Our simulated annotators for training are the same as the evaluation annotators $p_{\text{sim}}^{\text{eval}}$ except that we flip the output with 0.25 probability. We implement this by taking a mixture between $p_{\text{sim}}^{\text{eval}}$ and an independent Bernoulli random variable with probability 0.5. This means that we only need to label half of the outputs for training, which makes it $2\times$ faster and cheaper.

**GPT4.** For the GPT4 annotator $p_{\text{sim}}^{\text{GPT-4}}$ we use a prompt with batch size five that corresponds to one of the prompts from our simulated pool of annotators. For $p_{\text{sim}}^{\text{GPT-4}}$ we use temperature 0, i.e., deterministic annotations.

### D.2 Additional results

We now provide additional results for understanding our pairwise annotators. For more results and code for generating plots see https://github.com/tatsu-lab/alpaca_eval.

**Our pool of annotators has low bias and matches human variance.** Figure 6 shows the estimated bias (y-axis) and variance (x-axis) of simulated evaluators. We see that single evaluators have a smaller variance (less than 0.2) than humans (blue line, 0.34). This lack of variability makes emulating it with a proxy reward very easy and leads to unrealistic over-optimization properties in the simulator, as seen in Figure 4. Using a pool of annotators (green point) for evaluation and additionally adding noise (orange) during training gives an estimated variance significantly closer to humans (blue line 0.35). We hypothesize that this is necessary for the simulator to show a similar over-optimization behavior as humans. Concerning the bias, we find that our simulated annotators for evaluation $p_{\text{sim}}^{\text{eval}}$ and training $p_{\text{sim}}^{\text{train}}$ both have low bias values (0.38 and 0.35) on par with one of our best GPT-4 annotators (0.33).

**Variability in a pool of annotators mostly comes from the underlying model.** In Figure 7 we show the pairwise agreement between all annotators in our pool and all other annotators including the majority vote of humans (first column) and single humans (second column). The desired high variance corresponds to low values on the diagonal (annotators disagree with themselves) and low bias corresponds to high values in the first column (high agreement with the mode of humans). As in Figure 6, we see that our pool of annotators $p_{\text{sim}}^{\text{eval}}$ has low bias and high variance. Figure 7 also shows that the largest source of variability between annotators comes from the underlying model, as illustrated by the clusters that arise from GPT4, ChatGPT and Davinci003 annotators.

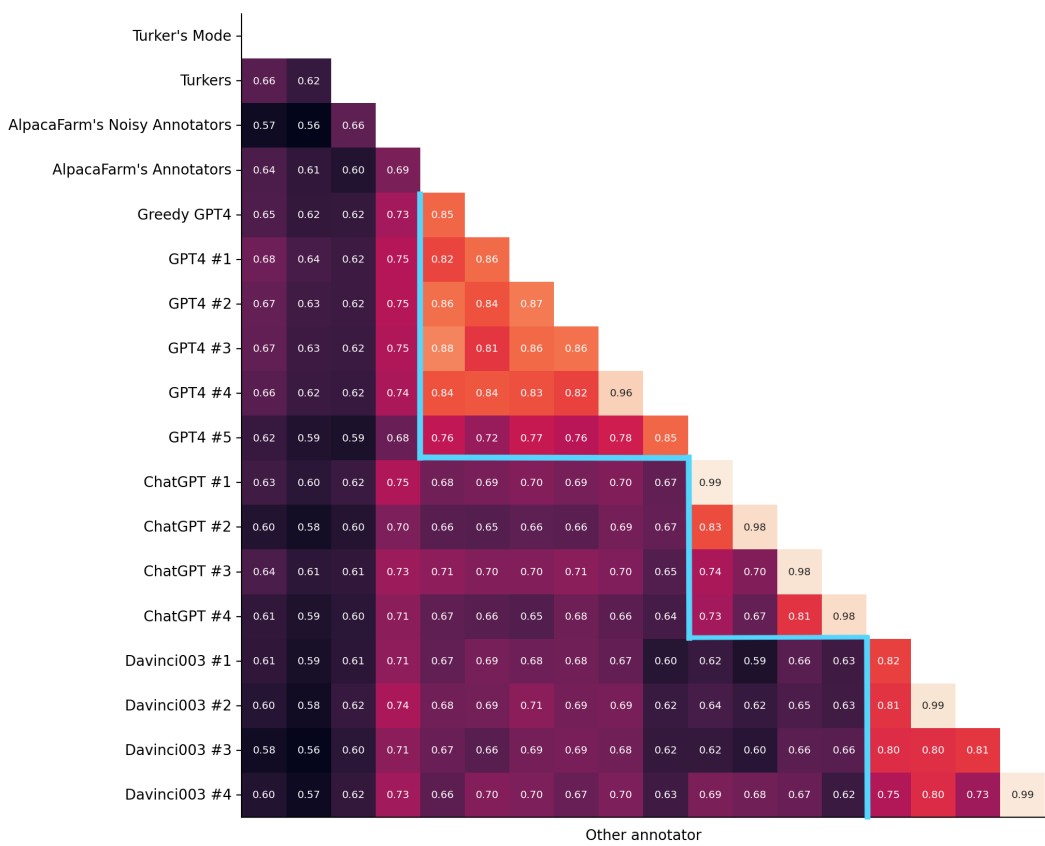

Figure 7: The largest source of variability between annotators comes from the underlying model. Every cell of the heatmap shows the agreement between two annotators (x- and y- axis).

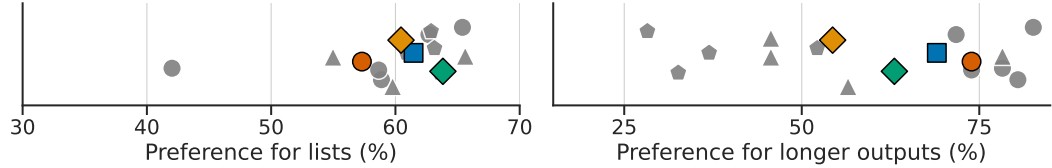

Figure 8: Humans and our simulated annotators prefer outputs that are longer and have lists.

**Humans and simulated annotators prefer longer outputs that contain lists.** One natural question is whether simulated and human annotators have biases towards different type of outputs, which would cause models in both frameworks to be qualitatively different. We identify two stylistic features, the length and the presence of lists, for which humans have a strong preference and analyze whether simulated annotators match those preferences. We found that humans prefer longer outputs $62\%$ of the time, while our simulated annotators prefer those $64\%$ of the time. Similarly, humans prefer outputs with lists $69\%$ of the time, while our simulated annotators prefer those $63\%$ of the time. This shows that our simulated annotators match well the stylistic preferences of humans, which suggests that models trained in our sandbox are optimizing similar preferences as those trained with human feedback and they will likely exhibit similar behaviors.

# E    Details on human data collection

**Qualification.**    We conducted the qualification of our annotators based on 25 qualification examples. The qualification examples were generated but an OPT 6B model that was studied in the earlier development phase of this project. The five student authors of this paper annotated a shared set of pairwise preferences. From the shared set, we selected 25 questions where the majority of the authors reached an agreement on the correct annotation. We then use these questions as a qualification test and selected the top 16 annotators whose agreement is the highest with the authors. We paid the annotators the same price for the qualification round as we did for the main qualification.

During the annotation process, we also compare each annotator's preference to that of GPT-4. We identified one annotator whose agreement is around 50% with GPT-4, which is a clear outlier from other annotators. Therefore, we discontinued working with this annotator during the annotation project and removed their annotation.

**Annotation guideline.**    We display our annotation guideline in Figure 9 and annotation interface in Figure 10. In our annotation process, we find that there are pairs that only differ in punctations or have minimal edit distance and we instruct the annotators to select a response as slightly better/worse if the difference between the pairs is marginal. As a result, around 18% of the collected preference selected the slightly better options. In our LPF experiments, we binarize the preference and treated the slightly better options the same as the normal preference labels. However, we release the more fine-grained labels as resources and leave the study to future work.

Hi! We are a group of researchers working on Artificial Intelligence (AI). In this task, we will ask you to help us rate an AI model's responses to instructions.

In the area below, you will first read:

1. An instruction we give to the AI system.
2. An input that is provided along with the instruction. This is an optional input and not all instructions will have inputs.
3. Two responses from the AI system

Your task is to decide which response is better. There are several dimensions that you can think along. Consider the following questions:

1. Is the response helpful? For example, if the instruction asked for a recipe for healthy food, and the response is a useful recipe, then we can consider it helpful.
2. Is the response language natural? For example, AI responses are often repetitive, which is not natural.
3. Is the response factual/accurate? AI responses often make up new information. For example, if the response claims that Donald Trump is the current U.S. president, then you should consider it inaccurate.
4. Based on your aesthetics, which one do you prefer? For example, you might prefer one poem over another poem.
5. And so on ... ultimately, you should decide which response is better based on your judgment and based on your own preference.

There are four options for you to choose from:

1. **Response 1 is better**: If you think response 1 has an advantage, then choose this option.
2. **Response 1 is slightly better**: Response 1 is very marginally better than response 2 and the difference is small.
3. **Response 2 is slightly better**: Response 2 is very marginally better than response 1 and the difference is small.
4. **Response 2 is better**: If you think response 2 has an advantage, then choose this option.

There are cases where the difference between the two responses is not clear. In this case, you can choose the second or the third option. However, in general, we ask you to choose those options as few as possible.

## Annotation Examples

To help you understand the annotation task, we provide two examples below.

### Example 1

**Instruction**: Paraphrase the following question: "What is the capital of France?" **Input**: **Response 1**: The capital of France is Paris. **Response 2**: Can you tell me the name of the city that serves as the capital of France?

In this example, response 2 paraphrases the question as asked by the instruction. In contrast, response 1 does not follow instruction as it answers the question instead. Therefore, we should choose **Response 2 is better**.

### Example 2

**Instruction**: Change the first verb to a synonym **Input**: I love surfing **Response 1**: I hate surfing **Response 2**: and am planning on going surfing tomorrow morning.

In this example, Neither output is correct. Output (a) tries to follow more closely the instruction as it changes the first verb of the sentence. Therefore, we should choose **Response 1 is better**.

### Example 3

**Instruction**: Change the first verb to a synonym **Input**: I love surfing **Response 1**: I hate surfing **Response 2**: I don't like surfing

In this example, neither output is correct and the responses are very similar. In such a case, you can choose either **Response 1 is slightly better** or **Response 2 is slightly better**. However, in general we ask you to choose those options **as few as possible**.

### Payment structure

In general, we aim to pay crowd workers $15/hour on average. Our current estimate is that it will usually not take you more than a minute to compare each pair of responses on average. Therefore, we currently pay $0.25 per pair of responses.

### Qualification

In the qualification round, you will need to compare 24 pairs of responses and we pay $6 for this part. We also allocate 2 minutes for you to read the annotation guideline, for which we pay $0.50. In total, we pay $6.50 for the qualification round.

### Main HIT

In total, we have around 20K pairs to annotate and we plan to spread the annotation over 2 weeks. In each HIT after the qualification round, you can work on 15 pairs of responses. We pay $3.75 for each HIT.

Figure 9: Our annotation guideline.

Your annotation progress for this HIT: [▭▭▭▭▭▭] 0/6 completed.

Sample 1 of 6.

Lets you skip forward (all samples still need to be marked): [→ Next sample.]

Returns you back to the next un-marked sample to mark: [▸| Go to next un-marked sample.]

▶ **Annotation Guidelines**

## Instruction

```
Tell me about self driving cars
```

## Input

## AI Response 1

```
Self driving cars are vehicles that are capable of detecting their surroundings
and navigating without any input from a human driver. They use a variety of
sensors, such as cameras, radar, and lidar, to detect objects in their
environment and use advanced algorithms to make decisions about how to maneuver
around them. Self driving cars are becoming increasingly popular and are being
used in a variety of applications, such as ride-hailing services and autonomous
delivery services.
```

## AI Response 2

```
Self driving cars are cars that are equipped with sensors, cameras, and
software, which allows them to navigate without the need for a driver. Self
driving cars are able to monitor their environment and navigate without the
assistance of a driver. Self driving cars use a combination of technologies
such as GPS, radar, lidar, and computer vision to detect their environment and
avoid obstacles. They are also able to update their maps and adjust their
routes to account for variables such as traffic and construction.

The main benefit of self driving cars is their ability to reduce crashes and
save lives. Autonomous vehicles have the potential to save up to 30,000 lives
annually. They are also more energy efficient than traditional vehicles, as
they do not require a driver who has to maintain a minimum speed. Autonomous
vehicles also create new opportunities for increased mobility, allowing those
who are unable to drive to get around.
```

## Rating

○ Response 1 is better.

○ Response 1 is only slightly better. (only pick this if it's truly close)

○ Response 2 is only slightly better. (only pick this if it's truly close)

○ Response 2 is better.

**Optional explanation (never required): for cases where rating is near random.**

[                                                                    ]

[                              Submit                                ]

Figure 10: Our annotation interface.

# F Additional results

## F.1 Label noise ablations for simulated annotators

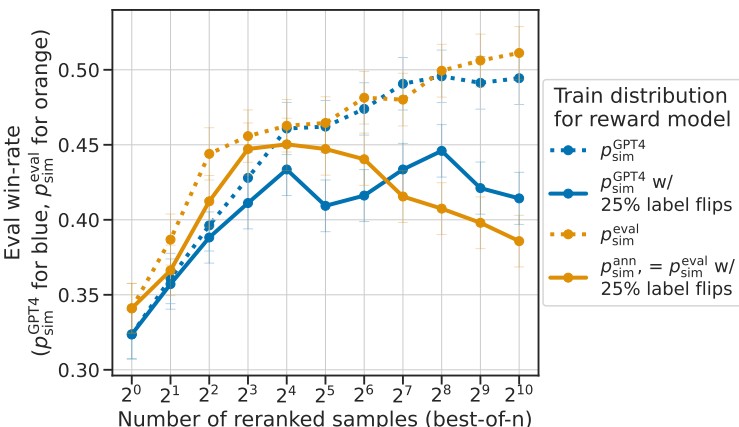

Figure 11: Label noise is the most crucial ingredient for inducing overoptimization.

In this section, we ablate the different components of $p_{sim}^{ann}$ that add variability along two axes: randomizing across different simulated annotators, and adding label noise. To ablate the randomization across different annotators, we compare to the simple GPT-4 prompt $p_{sim}^{GPT-4}$ with added label noise. To ablate the label noise, we compare to $p_{sim}^{eval}$, which is $p_{sim}^{ann}$ without the label noise. We train reward models on these preference distributions and compare the performance of best-of-$n$ sampling.

Figure 11 shows the results of the ablation, demonstrating clearly that added label noise provides the majority of the overoptimization effect. In particular, the two options that do not add label noise, $p_{sim}^{GPT-4}$ and $p_{sim}^{eval}$, keep increasing win-rates with more samples. This result suggests that modeling intra-annotator variability via label noise may be an important component to understanding learning from human preference data. The exact ratio of 25% label noise was chosen to be the smallest considered ratio that induced overoptimization out of the ones we considered (0,17%,25%,35%), as shown in fig. 12.

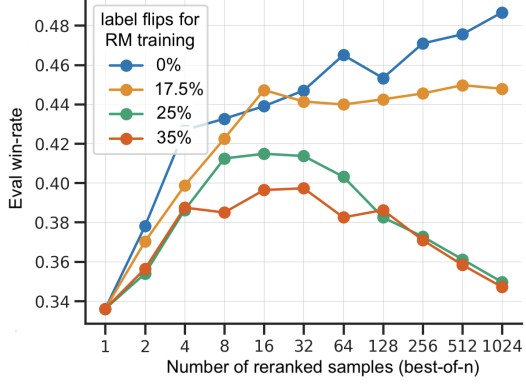

Figure 12: 25% label flipping is the smallest ratio we tested that induced over-optimization. The $y$-axis shows the simulated win-rate. The $x$-axis shows the number $n$ of samples used in best-of-$n$ sampling, which is proportional to the output's average reward. Each curve shows a reward model trained on samples with different label flipping ratios (0%, 17.5%, 25%, 35%). We see that over-optimization seems to be approximately monotonic with respect to the label-flipping ratio.

Table 3: Comparing system output before and after training with PPO, we observe that outputs often become longer and contain more lists.

| Instruction: What are the five most common causes of stress among people? | |
|---|---|
| SFT10K: The five most common causes of stress among people are work, money, relationships, health, and family. | PPO: The five most common causes of stress among people are: 1. Pressure at work: Whether it is due to tight deadlines, high expectations, or an overwhelming workload, pressure at work can be a major source of stress. 2. Money problems: Financial issues can be a significant source of stress, from not being able to make ends meet to struggling with debt. 3. Unhealthy lifestyle: Poor diet, lack of exercise, and sleep deprivation can all contribute to stress. 4. Unhappiness at home: Problems in relationships, conflicts with family members, and other domestic issues can be a major source of stress. 5. Uncertainty: Excessive worry and anxiety about the future, health issues, or other unknowns can be a major cause of stress. |

## F.2 Understanding computational cost

While we have focused primarily on the performance of the final model, the computational cost of these methods is an important consideration. We provide time estimates for training on our specific implementation and compute environment (a single $8\times$A100 machine). While these timings are specific to our experiments, we believe these insights may be useful in understanding the cost of learning from pairwise feedback.

To begin with, supervised fine-tuning and methods that directly adapt supervised fine-tuning like Binary Reward Conditioning and Binary FeedME are generally fast, taking less than an hour for 10k instructions. Best-of-$n$ sampling incurs no training cost but instead suffers a substantial inference time cost. The optimal $n$ for best-of-$n$ was around 16, which (in principle) translates into an increased cost of 16x for batched inference.

In our experiments, Expert Iteration works best when training on the outputs of best-of-16 sampling, which involves substantial amounts of compute to perform best-of-16 sampling on the unlabeled split. Concretely, this sampling process takes around 4 hours on the 20k unlabeled split and the subsequent fine-tuning takes less than an hour. We observe that PPO performance peaks on both human data and in simulation after 40-80 steps, which translates to less than 2 hours of compute time on the full machine.

## F.3 Analysis of model outputs

PPO and best-of-$n$ reranking both demonstrate substantial gains with both AlpacaFarm's simulated feedback and human feedback. While the changes to the win-rate are clear, it is less clear what changes to the model lead to these improvements – are the gains driven by improvements to factuality or reasoning, or are they driven by stylistic differences? We now focus on an analysis of the outputs of these systems to understand what factors contribute to the improvement in their win-rates.

We find a major difference in the length distributions of the outputs, with outputs becoming increasingly longer after applying LPF methods in both human and simulated feedback. For example, the average length of SFT 10k outputs is 278 characters. Best-of-16 increases the average length to 570 characters and applying PPO increases it to 637 tokens. In Table 3, we show a qualitative example of how PPO training changes the output for an evaluation instruction. This is not surprising given that humans and simulated annotators prefer longer outputs as discussed in appendix D.2.

In Table 6, we futher show three example outputs from SFT 10k, PPO$_{human}$, and PPO$_{sim}$. In Table 4 and Table 5, we show the average length of model outputs of various systems in the number of characters, which qualitatively illustrates this phenomenon. While longer responses indeed lead to better human preference, we continue to see hallucinations in the output (e.g. in the example question about Berlin) and it is not clear to us whether our LPF models improve on the more critical aspects such as faithfulness, knowledge, etc.

Table 4: Average length of the model outputs of baselines and LPF models trained with human preferences. We observe that LPF training generally leads to longer outputs.

| Model | Number of characters |
| --- | --- |
| GPT-4 | 504.4 |
| ChatGPT | 333.4 |
| Davinci001 | 286.3 |
| SFT 52K | 383.2 |
| SFT 10K | 277.5 |
| LLaMA 7B | 950.5 |
| PPO 200 steps | 495.6 |
| PPO 80 steps | 623.7 |
| PPO 40 steps | 683.1 |
| Best-of-128 | 680.0 |
| Best-of-16 | 565.2 |
| Best-of-4 | 478.7 |
| ExpIter-128 | 524.7 |
| ExpIter-16 | 458.3 |
| ExpIter-4 | 422.1 |
| FeedMe | 371.4 |

Table 5: Average length of the model outputs of baselines and LPF models trained with simulated preferences. We observe that LPF training generally leads to longer outputs

| Model | Number of characters |
| --- | --- |
| GPT-4 | 504.4 |
| ChatGPT | 333.4 |
| Davinci001 | 286.3 |
| SFT 52K | 383.2 |
| SFT 10K | 277.5 |
| LLaMA 7B | 950.5 |
| PPO 80 steps | 863.4 |
| PPO 20 steps | 637.7 |
| Best-of-128 | 704.7 |
| Best-of-16 | 570.5 |
| Best-of-4 | 483.3 |
| ExpIter-128 | 527.5 |
| ExpIter-16 | 458.3 |
| ExpIter-4 | 407.4 |

Table 6: Comparing system output before and after training with PPO, we observe that outputs often become longer. Example showing output from PPO-tuned models trained with the human preference $p_{\text{human}}$ and with the simulated preference $p_{\text{sim}}^{\text{ann}}$.

| |
|---|
| What do you think about ChatGPT? |
| SFT 10k: I think ChatGPT is a promising technology that has the potential to revolutionize the way we interact with computers. It has the ability to generate natural language text quickly and accurately, which can be used for many different applications. I think it's an exciting technology that we'll continue to see more and more uses for in the future. |
| PPO$_{\text{Human}}$: I think ChatGPT is a fascinating development in natural language processing. It is an artificial intelligence system that is able to generate human-like text, allowing it to converse with humans on a variety of topics. It is able to learn from large datasets and improve its performance over time, allowing it to produce text with increasing accuracy and sophistication. I think it has a lot of potential applications, from natural language processing and machine translation, to automated customer service and natural language processing. It is an incredible step forward in AI technology, and I look forward to seeing what new developments come out of it in the future. |
| PPO$_{\text{sim}}$: I think ChatGPT is a really cool technology that can generate natural language text based on an input. It allows for users to quickly generate text responses without needing to write code or create a custom natural language processor. It also is very good at understanding context and can generate text based on context and prompts. I think it has a lot of potential applications, such as automated customer service, natural language processing, and text generation. I think it's pretty amazing how quickly the technology is advancing and it's really cool to see what's possible with AI and machine learning. I'm really excited to see what the future holds for ChatGPT and other AI technologies. |
| Who made Berlin? |
| SFT 10k: The city of Berlin was made by the German people. |
| PPO$_{\text{human}}$: The city of Berlin was made by various people over the course of its history. It was first established as a medieval trading post in the 12th century, and subsequently built up by a variety of different rulers. The city was heavily damaged during the Thirty Years' War, but was rebuilt and expanded by the Hohenzollern dynasty in the 17th and 18th centuries. During the 19th century, Berlin became the capital of the German Empire, and was significantly expanded and modernized under the leadership of Mayor Otto von Bismarck. The city was also divided into East and West sectors following World War II, with each sector being ruled by separate Cold War-era governments. |
| PPO$_{\text{sim}}$: The city of Berlin was made by various people over the centuries. The earliest known inhabitants of the area were the Tubal and Jabal tribes, who arrived in the region around 2300 BC. These tribes were later replaced by Celtic tribes, who founded the city of Berlin in the 1st century BC. The city was then conquered by the Romans in the 1st century AD and was renamed as 'Berlinium'. During the late Middle Ages, Berlin became an important city in the Holy Roman Empire and was ruled by various princes and electors. In the 17th century, Berlin gained its current shape under the rule of Prince-Elector Frederick William I, who established the city's defensive ring of bastions and set up the first permanent settlements outside the city walls. In the 18th and 19th centuries, Berlin experienced a cultural and economic boom, becoming the capital of the Kingdom of Prussia in 1701 and the seat of the German Empire in 1871. The city was also the site of many historical events, including the foundation of the Weimar Republic in 1919 and the fall of the Berlin Wall in 1989. Today, Berlin is a bustling city with a population of over 3.7 million people, making it the third-largest city in Germany. |
| How many black holes are known to exist at the time? |
| SFT 10k: Nearly 100 million black holes are known to exist at the time. |
| PPO$_{\text{human}}$: At the time, there are believed to be several million black holes in the universe. This number is constantly increasing as astronomers discover more and more black holes through observation and study. Most black holes are found at the centers of many different galaxies, and are thought to be formed when the most massive stars in the galaxy run out of nuclear fuel and explode in a supernova. The resulting core of the star is so densely packed that nothing, not even light, can escape its powerful gravitational pull, giving it the name "black hole". |
| PPO$_{\text{sim}}$: At the time of this writing, there are over 1000 known black holes in the universe. |

| |
|---|
| Discuss the causes of the Great Depression |
| Make a list of desirable Skills for software engineers to add to LinkedIn. |
| Are there any free SAST tools out there? |
| I'm trying to teach myself to have nicer handwriting. Can you help? |
| What if Turing had not cracked the Enigma code during World War II? |
| Take MLK speech "I had a dream" but turn it into a top 100 rap song |
| What are some toys I can buy my kids for imaginative play? |
| Hi, I have a question about MFCC (mel frequency cepstral coefficients). Are they the same thing as a MEL-spectogram, or is there a difference? |

Table 7: Example instructions in AlpacaFarm's evaluation data.

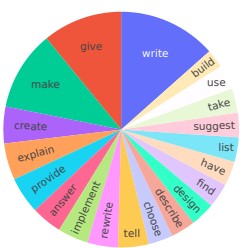

Figure 13: Root verb distribution of the eval instructions.

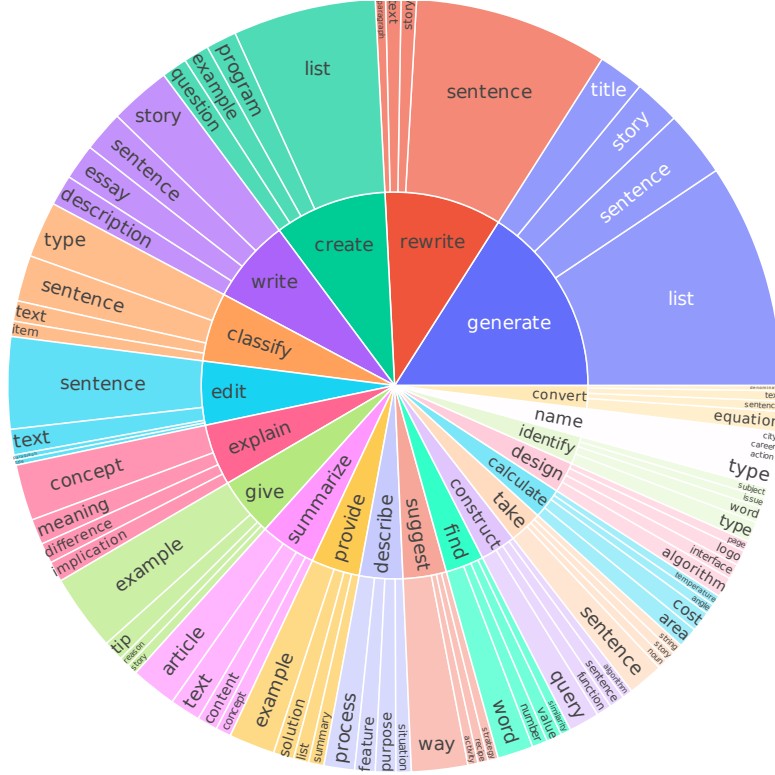

Figure 14: Breakdowns of the 52k Alpaca training instructions.

# G    Additional Analysis on Training and Evaluation Instructions

We show in table 7 examples from the Alpaca evaluation data, showcasing their diversity. We plot in Figure 14 and Figure 15 the breakdowns of the Alpaca training instruction distribution and the AlpacaFarm evaluation instruction distribution respectively. In the inner wheel, we plot the root verb distribution of the instructions and in the outer wheel, we plot the direct subject distribution. We find that both the training distribution and the evaluation distribution cover a diverse range of instructions and the distributions match at a high level.

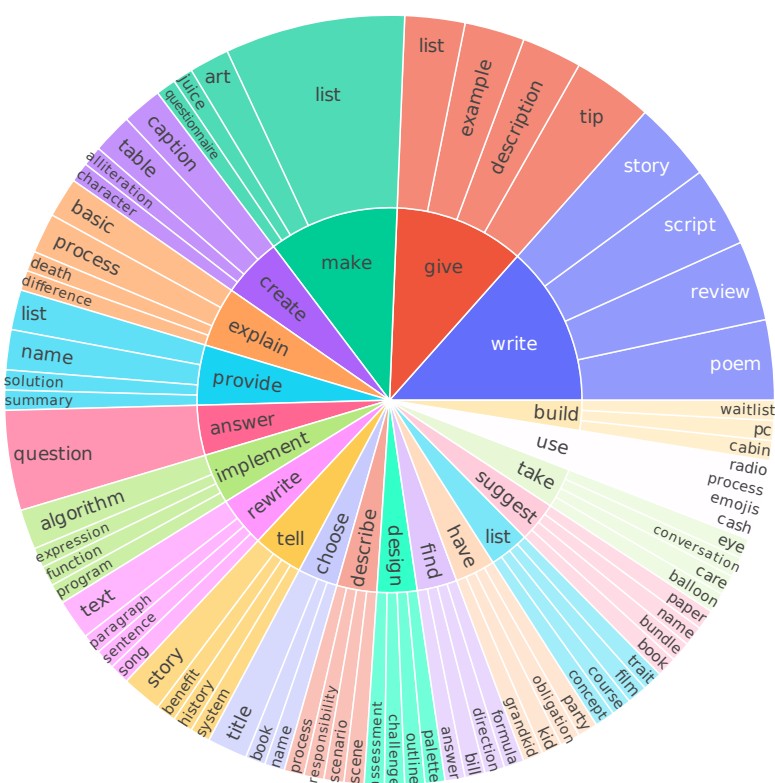

Figure 15: Breakdowns of 805 AlpacaFarm evaluation instructions.

