# A Reference LPF methods on AlpacaFarm

Having defined and validated the pairwise feedback simulator and evaluations in AlpacaFarm, we now turn our attention to studying methods that learn from pairwise feedback on AlpacaFarm. Unfortunately, the lack of existing benchmarks for learning from pairwise feedback for instruction following means that there has not been any open study of these methods in the instruction-following setting. To address this, we implement a suite of reference methods and benchmark them on AlpacaFarm.

In the remainder of this section, we will introduce our reference methods, which fall into two categories based on whether they fit a surrogate reward model as part of the learning process.

## A.1 Methods that directly learn from pairwise feedback

**Binary FeedME.** FeedME is a method proposed by OpenAI [45] that incorporates human feedback with supervised fine-tuning on model generations that are rated 7/7 by human labelers. We adapt this approach to the pairwise feedback setting and call this baseline binary FeedME. This approach fine-tunes the SFT model on the chosen response in each preference pair with supervised learning.

**Binary Reward Conditioning.** Motivated by controllable generation through conditioning [27, 34, 29, 21], we propose binary reward conditioning, a baseline method that fine-tunes the SFT model with the feedback data $\mathcal{D}_{\text{pairwise}}$ by conditioning instances with either a positive or negative control token. Specifically, for each instance $(x, y_0, y_1, z) \in \mathcal{D}_{\text{pairwise}}$, the string concatenation of instruction $x$ and response $y_z$ denoted as $[x, y_z]$ is prepended with the positive token and used in supervised fine-tuning (similarly $[x, y_{1-z}]$ is prepended with the negative token). This process creates a modified demonstration dataset that is double the size of $\mathcal{D}_{\text{pairwise}}$. At test time, we draw samples from the fine-tuned model conditioned on the positive token.

## A.2 Methods that optimize a surrogate reward function

We now describe methods that incorporate feedback by first building a surrogate reward model with pairwise feedback data. To start, we describe the step of training the surrogate reward model.

To train a parameterized surrogate $\hat{R}_\phi$, one can maximize the log-likelihood of the preferences $z$ under the Bradley-Terry model [10]

$$\text{maximize}_\phi \sum_j \log P(z^{(j)} \mid x^{(j)}, y_0^{(j)}, y_1^{(j)}) = \sum_j \log \frac{\exp(\hat{R}_\phi(x^{(j)}, y_z^{(j)}))}{\exp(\hat{R}_\phi(x^{(j)}, y_0^{(j)})) + \exp(\hat{R}_\phi(x^{(j)}, y_1^{(j)}))}. \tag{1}$$

Once the surrogate reward model is trained, both training and inference algorithms can optimize against the reward model rather than query pairwise feedback. While this can be a powerful approach, we will see that it can also lead to *over-optimization* [19] where models learn to exploit the reward model rather than achieve high true reward. We now describe 4 methods that leverage the surrogate reward model.

**Best-of-$n$ Sampling.** Best-of-$n$ sampling (or re-ranking) [64, 5, 22, 8] is a common inference-time method that aims to improve the generation quality. Given an input $x$, the method returns the response with the highest surrogate reward value among $n$ i.i.d. responses drawn from the SFT model. While simple to implement and useful as a baseline, this approach incurs high inference costs.

**Expert Iteration.** Expert iteration [2, 61, 71] is a technique that has recently been used to train language models. We adapt this approach in AlpacaFarm as a two-step method. In the first step, we perform best-of-$n$ sampling and store the generated samples. In the second step, we fine-tune $p_{\text{SFT}}$ on these samples with supervised learning. While prior work applies expert iteration for multiple rounds by performing best-of-$n$ sampling again for intermediate models, we focus on performing a single round. In Appendix F, we include our preliminary study of multi-round expert iteration.

**Proximal Policy Optimization.** Proximal Policy Optimization [PPO; 25, 59] is a popular RL algorithm that has been recently used to develop InstructGPT [47] and ChatGPT [44]. When applied

to fine-tune LMs with RLHF, PPO maximizes the following KL-regularized objective w.r.t. model parameters $\theta$

$$\mathbb{E}_{x\sim p(x),\ y\sim p_\theta(y|x)}\left[\hat{R}_\phi(x,y) - \beta\log\frac{p_\theta(y\mid x)}{p_{\text{SFT}}(y\mid x)}\right],\tag{2}$$

where $p(x)$ is an unlabeled instruction distribution, $p_\theta(y\mid x)$ is fine-tuned from the $p_{\text{SFT}}$ model, and $\beta\in\mathbb{R}$ is a regularization coefficient. Each step of PPO alternates between drawing samples from the current policy and performing gradient updates based on the pool of samples with importance sampling and clipping. We follow existing PPO implementations and document the modifications we made in Appendix F.

**Quark.** Quark is inspired by reward conditioning and has been shown to be effective for controllable generation tasks. Like binary reward conditioning, Quark on train sequences with prepended control tokens. Unlike binary reward conditioning, Quark bins model samples into multiple groups based on the reward value, adds KL and entropy regularization, and repeats the entire process across multiple rounds.

In our preliminary analysis, we find the top-quantile variant reported in [38], i.e. only training on the best reward group, to perform the better than the all-quantiles variant which trains on all groups. We document our tuning process and implementation modifications in Appendix F.

# B   Tradeoffs in simulator design

We have now seen that insights on methods development in the simulator successfully transfer to real human data. To do so, pairwise feedback in the Farm relies on two important design decisions: the use of multiple API models with different prompts, and injecting label noise into the training preferences. Though these components were designed to better simulate humans, who exhibit a high level of inter-annotator disagreement, it is reasonable to assume that each of these components degrade the quality of the resulting training data. Specifically, both Davinci003 and ChatGPT are worse pairwise evaluators than GPT-4 (Figure 5), and added label noise further degrades the training signal. One way to analyze whether these design decisions are necessary is by comparing to a simple yet high-quality baseline: a single-prompt GPT-4 (red in Figure 5). Specifically, in the remainder of this section, we answer the two following questions:

- Does modeling variability in preferences result in a better simulator over the simple GPT-4 alternative?
- How much does the variability impact the performance of the trained model outside Alpaca-Farm?

## B.1   Comparing simulators to reality

To answer the question on simulator design, we first analyze learning dynamics on real human preference data, as shown in Figure 3 left. We look at the three most effective methods in our suite, all of which optimize against a fixed reward model trained on human preferences.

Figure 3 left displays the tell-tale sign of *reward model over-optimization*, the phenomenon where optimizing against a fixed reward model only increases model performance up to a point. Noted in prior work [19], this effect is the result of the human reward proxy model $\hat{R}_\phi$ being an imperfect estimate of the (unknown) human reward $R$. This effect is most evident for PPO (shown with steps $40, 80, 200$) and Best-of-$n$ ($n = 4, 16, 128$). Expert Iteration (shown with $n = 4, 16, 128$) plateaus and does not over-optimize yet, which may be tied to its inability to optimize the reward model reward effectively.

We find exactly this result with the simulator. Figure 3 middle shows the analogous plot for each of the methods trained in the AlpacaFarm. We see a similar story of over-optimization: PPO (steps 20 to 200) and Best-of-$n$ ($n = 1$ to 1024) show a clear increase then decrease in win rate, with Expert Iteration ($n = 4$ to 128) plateauing in performance.

This over-optimization behavior is critically linked to our choice of modeling annotator variability in the simulator. As an alternative, we re-trained all the models using a single high-accuracy prompt for

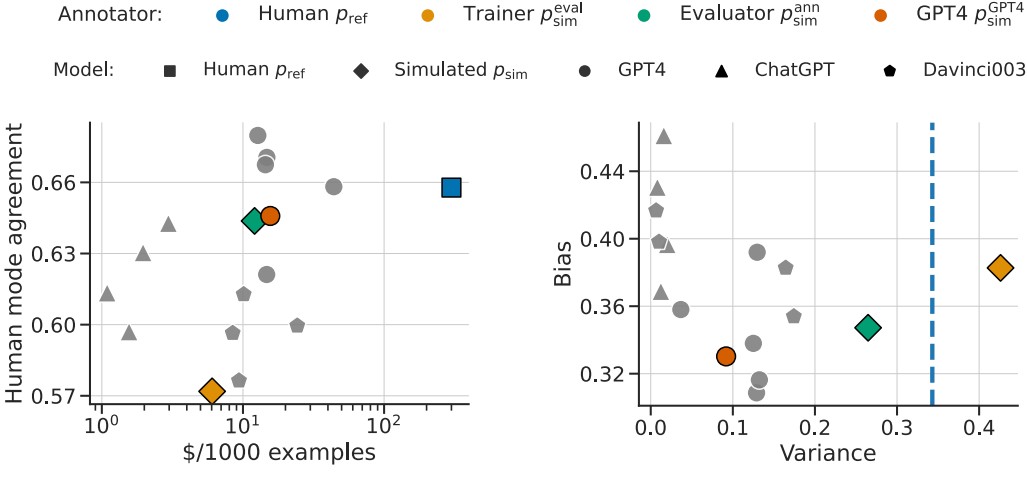

(a) Price and performance of annotators

(b) Bias and variance of annotators

Figure 5: Our simulated annotators are cheap and match well with human annotators. Grey points are all the annotators in our simulated pool, the green point shows the resulting pool of annotators (which we use for evaluation), the orange point shows the same simulated pool with additional noise (which we use for training), the blue point shows the pool of human annotators, and the red point shows a single low variance GPT-4 annotator which we analyze in appendix B. (a) shows the price (x-axis) and the performance (y-axis) of annotators as measured by their agreement with the mode of human annotations. (b) shows the estimated bias and variance of simulated annotators compared to humans, whose bias is by definition 0, and variance is shown with a blue line.

GPT-4, $p_{\text{sim}}^{\text{GPT-4}}$. The prompt was chosen as it has a high correlation with human annotators (red in Figure 5) while still being deterministic (preferences were sampled with greedy decoding, temp=0).

The results are shown in Figure 3 right, which displays a clear departure from the over-optimization result in the other two figures. The win rate of all three methods – Best-of-$n$, PPO, and Expert Iteration – continues to increase with compute and shows little signs of plateauing. While this lack of over-optimization may appear to be a benefit, it poses a major challenge for reliable methods development. In a simulator without over-optimization, picking the best method would simply be decided by which method was dedicated more compute. For example, Figure 3 right shows Best-of-1024 to be much better than PPO, which is a reversal of the trend observed on human data.

Returning back to the first question asked in this section, these results make it clear that the design of pairwise preferences in AlpacaFarm is necessary to create a simulator that closely matches reality. In Appendix E, we ablate the design more finely and find that the added label noise provides the majority of the benefit for inducing over-optimization.

## B.2 Transfer properties of AlpacaFarm trained models

Finally, we have shown that replicating annotator variability is important to capture over-optimization behaviors in AlpacaFarm. However, we highlight that the single GPT-4 prompt annotator can be useful when the goal is to directly transfer models from AlpacaFarm to the real world (rather than re-training with human preferences).

To illustrate this point, we compare a PPO model (step 40) trained in the Farm (on $p_{\text{sim}}^{\text{ann}}$) with a PPO model (step 30) trained on the single-prompt GPT-4 (on $p_{\text{sim}}^{\text{GPT-4}}$). We then measure their win rate according to human preference evaluation, $p_{\text{human}}$.

We find that $\text{PPO}_{\text{sim}}^{\text{ann}}$ trained in the Farm achieves a win rate of 43%, while $\text{PPO}_{\text{sim}}^{\text{GPT-4}}$ trained on single-prompt GPT-4 data achieves a win rate of 50%. To contextualize these results, the initial SFT model has a win rate of 44%, $\text{PPO}_{\text{human}}$ has a win rate of 55%, and the best non-PPO human method has a win rate of 51% (Best-of-16) (full table in Appendix E).

These results demonstrate a strong tradeoff in simulator design: more faithful simulators which display greater over-optimization train objectively worse models. The standard AlpacaFarm pairwise evaluators are suitable and ideal for developing new methods and comparing to others, as Figure 2 demonstrates that these comparisons transfer when the methods are re-trained on human data. However, for training new models in the simulator and deploying directly thereafter, a single consistent annotator such as $p_{\text{sim}}^{\text{GPT-4}}$ can provide significant gains on real-world evaluation.

## C   Related work

**Instruction following.**   Many works on instruction following focused on cross-task generalization, where models were trained on a pool of NLP tasks, and evaluated on other held-out tasks based on simple metrics [42, 75, 54, 4, 74]. Increasingly, instruction following methods have focused on real world human interaction patterns [47, 6], rather than collections of existing NLP benchmarks. For example, InstructGPT was developed on user instructions submitted to OpenAI API [47]. Our work builds upon these works by attempting to bridge the gap between the ease of development and evaluation of traditional academic benchmarks and the more complex algorithms and real-world settings of recent works on instruction following.

**Simulating human feedback.**   Constitutional AI [7] simulates human feedback with AI feedback for model development to improve harmlessness and helpfulness. AlpacaFarm, on the other hand, simulates human feedback with API LLMs so that simulated experiments reflect the outcomes of experiments performed with real human feedback. Due to the difference in goals, the construction and usage of the feedback simulator are different in the two settings. For example, AlpacaFarm's simulator perturbs LLM preferences with bit-flip label noise to mimic the noisiness of human annotation, whereas Constitutional AI's simulator doesn't inject extra noise.

The evaluation aspects of our work are related to a growing line of work on simulating human annotation for evaluation [16, 52, 15, 51, 35]. Our core evaluation and feedback mechanism makes use of the same underlying ideas, but our work is distinguished by a focus on using pairwise feedback for training, as well as careful validation beyond per-example agreement metrics. AlpacaFarm shows that LLM feedback can capture method-level correlations as well as qualitatively important features of human annotation, such as stylistic preferences and variability.

Our goal of emulating human annotators also connects to work on simulating humans with LMs based on personas [49, 48, 1, 3], as well as works that simulate human behavior in the context of cognitive science, social science, and economics [70, 26]. Our work complements these works by showing that simulated LLM annotators can replicate many of the qualitative features of training on pairwise human feedback.

More broadly, building a simulator environment to enable low-cost experimentation is common in the field of reinforcement learning and robotics [11, 68, 67, 65, 17, 24, 18]. Our work shares the same underlying motivations, but instead of simulating physical systems, AlpacaFarm simulates human preference feedback.

**Methods for learning from feedback.**   To hold annotation cost constant across learning methods, we have focused only on methods that learn from pairwise feedback in this work. However, there exist methods in the literature other than those explored in Section A that can incorporate alternative sources of feedback such as natural language [76, 33, 23, 60, 55, 13, 56, 41], numeric ratings [45, 32], or execution traces [14]. We view extensions of AlpacaFarm to these settings as exciting future work.

We have included a set of RL algorithms in our study that optimize the surrogate reward, but this set is by no means comprehensive. RL research applied to NLP has a long history [77, 63, 28, 50, 43, 31, 30, 53, 62], and we expect future work in this direction to benefit from the ideas and artifacts in AlpacaFarm.

## D   Limitations and future directions

**Difference between GPT4 and human feedback.**   Our sandbox assumes that LLMs can simulate human feedback. Section 4 shows that this assumption holds in the sense that LLM annotations align with the mode of human preferences and replicate its many features. However, we also observe

| |
|---|
| Discuss the causes of the Great Depression |
| Make a list of desirable Skills for software engineers to add to LinkedIn. |
| Are there any free SAST tools out there? |
| I'm trying to teach myself to have nicer handwriting. Can you help? |
| What if Turing had not cracked the Enigma code during World War II? |
| Take MLK speech "I had a dream" but turn it into a top 100 rap song |
| What are some toys I can buy my kids for imaginative play? |
| Hi, I have a question about MFCC (mel frequency cepstral coefficients). Are they the same thing as a MEL-spectogram, or is there a difference? |

Table 2: Example instructions in AlpacaFarm's evaluation data.

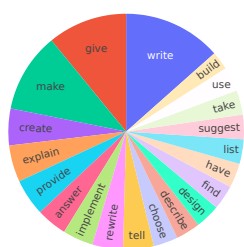

Figure 6: Root verb distribution of the eval instructions.

that no single LLM-based annotator captures the heterogeneity of human annotation, and substantial amounts of noise had to be injected in the simulated preference for rankings of methods trained in AlpacaFarm to match those trained with real human feedback.

In addition, we found that suitable hyperparameters for learning algorithms can be different for training with simulated feedback compared to human feedback. For example, due to changes in the scale of values of the surrogate reward model, the range of suitable KL regularization coefficients for RLHF is different.

Finally, we note that our preference simulator is validated with respect to our recruited pool of crowdworkers, and mimics their preference variability as a result. The best prompts for transferring insights to other, more consistent crowdworker pools may involve lower levels of noise or a smaller prompt ensemble.

**Limitation in current experiments.** Due to resource constraints, we have not conducted extensive hyperparameter tuning for the methods we studied in Section A. For each method, we have performed at least 3 tuning runs, and we detail our tuning process in Appendix F. It is possible that with more careful tuning, the comparison between PPO and alternative approaches like expert iteration and Quark might change.

Our study focuses on comparisons of methods that learn from pairwise feedback. We do not study other forms of feedback, such as numerical ratings or natural language. Neither do we study quality-quantity tradeoffs for learning from pairwise feedback. We leave this exposition to future work.

**Future directions.** We showed that AlpacaFarm substantially lowers the cost and iteration time of research on and development of methods for learning with pairwise feedback. AlpacaFarm provides a blueprint for constructing other useful simulators for AI research that requires human supervision, and we view it as an exciting opportunity to expand this simulation approach to support data from other domains as well as methods that learn from alternative forms of human feedback.

# E   Additional results

## E.1   Analysis of instruction data