# OpenReview forum: "AlpacaFarm: A Simulation Framework for Methods that Learn from Human Feedback"
_NeurIPS.cc/2023/Conference — NeurIPS 2023 spotlight_

### Official Review · Reviewer_NPeD · 2023-06-29

**Soundness:** 3 good
**Presentation:** 3 good
**Contribution:** 3 good
**Rating:** 7
**Confidence:** 4

**Summary:**

This paper presents a simulation framework, AlpacaFarm, for developing LLMs with human feedback.
AlpacaFarm adopts LLMs (e.g., GPT-4) to generate feedback (i.e., the ranking of candidate responses given the query), and evaluate the performance by calculating the win-rate against the baseline.
This framework can obtain synthetic feedback data cheaply compared with human annotators.

The major contribution is the system design and validation of the AlpacaFarm framework.

System Design:
- i) the framework constructs a relatively comprehensive evaluation dataset.
- ii) multiple LLMs are adopted as annotators proxy (GPT-4, ChatGPT and Davinci003)
- iii) random noise is injected to mimic the variability of human annotators.

Validation: Simulated evaluation  (i.e., the model performance ranking ) results of learning with the synthetic feedback dataset match that obtained by training models with human feedback and evaluated by real annotators. Besides, the pairwise ranking and variability also correlate with the human results well.

Finally, the paper benchmarks the reference methods with the framework of LLAMA-7B, suggesting that SFT plays a significant role and highlights the superiority of PPO.



**Strengths:**

- Clarity: The overall presentation is good, clearly conveying the key ideas of this paper.

- Significance: This framework would potentially become a useful resource for building, and evaluating the LLMs with feedback datasets when budgets are limited.

- Originality: The novelty of the framework is somewhat limited as generating synthetic feedback datasets & evaluating with powerful LLMs have already been explored recently. Yet, it is still worthwhile to explore combing these dots and justify its effectiveness.



**Weaknesses:**

- Some hyper-parameter choices are unjustified (such as the ratio in Question 1).
- The effect of this framework is under-explored when the baseline model is close to the annotators LLMs.

---
After Rebuttal:

Thanks for the response which well-addressed my questions well. I have raised my score to Accept.

**Questions:**

- The paper mentioned that 25\%  of the simulated preferences are randomly flipped. Any explanation on the ratio? Besides, while the results are consistent with the human results, I think this randomness may not faithfully reflect the inherent variance as human variability depends on the background and geography of the annotators.

- What results would be when the base LLM is close to the annotator LLMs? Will this framework still effective when we are  adopting a powerful LLM itself as annotators?

- What prompt is used to generate feedback and evaluate the candidate responses? Did you do some stabilization techniques such as self-consistency in [1] to stable the evaluation results?

- Typo:  Captions Table 4 and Table 5 of the Appendix: longer outputs.s ->  longer outputs.

[1] Large Language Models are not Fair Evaluators

---

> ### Author Rebuttal · Authors · 2023-08-08
>
> We thank the reviewer for their useful review, which we have incorporated to improve our paper.
>
> # Simulated annotators
>
> > *The paper mentioned that 25% of the simulated preferences are randomly flipped. Any explanation on the ratio?
>
> We have expanded our explanation in Appendix B.1 about the 25% ratio. We refer the reviewer to the [general rebuttal](https://openreview.net/forum?id=4hturzLcKX&noteId=Hq0kpzSHSw).*
>
> ----
> >*randomly flipped [...] while the results are consistent with the human results, I think this randomness may not faithfully reflect the inherent variance as human variability depends on the background and geography of the annotators.*
>
> As discussed in section 3.2 (text quoted below from line 153), there are two important sources of variability for human annotators: intra- and inter-annotator.
>
> >> To more completely emulate human annotators, we first emulate inter-annotator variability into the simulated pairwise preference by mimicking a pool of annotators. We design different annotators by querying different API LLMs and varying the prompts with different formats, batch sizes, and in-context examples.[...]. To emulate intra-annotator variability, we directly inject random noise and flip the simulated preference 25% of the time.
>
> The random flipping concerns intra-annotator variability, while using different API LLMs with varying in-context examples addresses the inter-annotator variability that the reviewer seems to be referencing. That said, we completely agree with the reviewer that our approach to emulating inter-annotator variability does not capture complex human variability, such as that due to background and geography. Our work is only a first step in this direction, and this important and challenging problem will require much future work to tackle. We have added this to our new limitations section. We refer the reviewer to the [general rebuttal](https://openreview.net/forum?id=4hturzLcKX&noteId=Hq0kpzSHSw).
>
> ---
>
> > *What prompt is used to generate feedback and evaluate the candidate responses? Did you do some stabilization techniques such as self-consistency in [1] to stable the evaluation results?*
>
> We thank the reviewer for their insightful comments. We detail our prompts in appendix G.1 and are releasing them as part of our code release. With respect to paper [1], we concur that the order of examples is crucial, as emphasized in the following excerpt from Appendix G.1:
>
> >>For each annotator, we randomize the ordering between the two outputs to annotate, i.e., we randomly choose which output is the first and which is the second. We found randomization to be important given that the first output is often preferred by simulated annotators
> Given this randomization we do not require the stabilization techniques form [1].

---

> > ### Comment · Reviewer_NPeD · 2023-08-16
> >
> > Thanks for your response and my questions are well-addressed. I will raise my score to Accept.

---

### Official Review · Reviewer_JtTg · 2023-07-05

**Soundness:** 3 good
**Presentation:** 3 good
**Contribution:** 4 excellent
**Rating:** 8
**Confidence:** 4

**Summary:**

This paper introduces AlpacaFarm, a simulator that enables faster and cheaper research and development of fine-tuning LLMs with human feedback. The authors propose to use an LLM to simulate human feedback, which is 45x cheaper than using crowdworkers and displays high agreement with humans. They also identify an evaluation dataset which is representative of real-world instructions from interactions between humans and chatbots or other LLMs. The authors also plan to open-source reference implementations for multiple methods which are widely used in the research and applications such as PPO, Best-of-N (BoN), and Expert Iteration (EI). The authors validate their simulator and evaluation protocols by showing  that rankings of models trained in AlpacaFarm match those of models trained on human data. The results indicate that leveraging a reward model can improve performance over supervised fine-tuning, which is in line with prior work.


**Strengths:**

## Clarity
The paper is well-written and clear overall. I also appreciated that the authors didn't only consider the agreement between LLMs and humans but also the variability in the answers which is an important aspect of human evaluation and feedback.

## Impact
The paper addresses a timely topic of great importance, namely understanding how to best fine-tune LLMs with human feedback. Although it is more of an engineering / datasets / benchmarks paper, I believe it fills a gap in the literature and could lower the entry barrier for doing research on RLHF and related methods which are rather complex and expensive to train well.

## Novelty
Although the paper does not propose a novel approach, it open-sources reliable implementations of popular approaches for fine-tuning LLMs with human feedback and proposes an automatic (fast and cheap) way of evaluating such methods. I believe these could be quite impactful and valuable for the community. I expect this can enable research and development of new methods, as well as a platform for fairly comparing them in order to make faster progress on these problems.

**Weaknesses:**

## Generality
One of the main weaknesses of this paper is the fact that only LLaMA-based models are considered. However, the LLM simulator is advertised as a general platform for researching, developing, and evaluating human feedback fine-tuning of LLMs. In particular, one open question is whether the models you use for evaluation and simulating human feedback transfer to other base models such as OPT, Pythia etc.

1) Thus, it would be useful to include experiments validating this hypothesis to demonstrate your protocol is robust to the base model used.

2) In addition, it would also be good to include different models for evaluating and even for simulating human feedback to see how much variance there is and increase the robustness of the results obtained with this platform. In particular, I expect that the results won't perfectly match human evaluation and feedback (particularly for the absolute performance), but adding different ways of simulating human feedback and evaluating LLMs can at least decrease the variance in the final results.

3) It would also be good if the authors can include human evaluations in the paper.

**Questions:**

1. In section 3.2, you claim that the agreement between your model and human annotators is high and quote 65%, which seems rather low to me. Can you provide more details on how this number was computed and what it means in practice? How does it compare with the agreement across humans?

2. Also in section 3.2, you mention that to emulate intra-annotator variability you inject random noise flipping the simulated preference 25% of the time. How did you come up with this number, did you run experiments to find out the intra-human variability? How did you validate that the resulting models match human variability?


**Limitations:**

The authors mention some of the assumptions and limitations of the paper but I strongly suggest having a separate section that discusses these in greater depth. Given the potential impact of this simulator and relevance to real-world applications, it is essential for readers to understand that this is just a simulator which may be inaccurate so it should only be used for research and preliminary experiments; human evaluations and potentially feedback should always be used before deployment or to make stronger claims about a model's capabilities.

---

> ### Author Rebuttal · Authors · 2023-08-08
>
> We thank the reviewer for their questions and will address their concerns in the updated manuscript.
>
> ## Limitations
> > *The authors mention some of the assumptions and limitations of the paper but I strongly suggest having a separate section that discusses these in greater depth.*
>
> We agree with the reviewer’s suggestion and have incorporated the feedback. We refer the reviewer to the [general rebuttal](https://openreview.net/forum?id=4hturzLcKX&noteId=Hq0kpzSHSw).
>
> ---
>
> > *one open question is whether the models you use for evaluation and simulating human feedback transfer to other base models such as OPT, Pythia etc.*
>
> This is an interesting and important question, which we now discuss in the new limitation section. During earlier stages of development, we experimented with OPT and Flan-T5 but found that outputs from these models were often completely wrong, making it challenging to obtain enough signal from human feedback. Since submitting our paper, many strong strong base models have been released (e.g. MPT, Falcon, LLaMA-2) and we think that it’s an important avenue for future work to consider the impact of the choice base model.
>
>
> ## Simulated annotators
>
> >  *you inject random noise flipping the simulated preference 25% of the time. How did you come up with this number, did you run experiments to find out the intra-human variability? How did you validate that the resulting models match human variability?*
>
> We thank the reviewer of their important questions, which we now expanded on in the paper. We refer the reviewer to the [general rebuttal](https://openreview.net/forum?id=4hturzLcKX&noteId=Hq0kpzSHSw).
>
> ---
>
> > *In section 3.2, you claim that the agreement between your model and human annotators is high and quote 65%, which seems rather low to me. Can you provide more details on how this number was computed and what it means in practice? How does it compare with the agreement across humans*
>
> As elaborated in section 4.3, the 65% is comparable to the human-human agreement rate at 66%. Quote from line 259:
>
> >> We begin computing agreement levels between our simulated annotator and a majority vote of 3 human annotators, comparing this to the agreement level of a held-out human annotator. We find that our evaluator $p_{eval}^{sim}$ has a 65% agreement rate with the human majority vote, which is similar to the held-out human agreement rate (66%) [...]
>
> We agree that this number seems low and see two reasons for that. First, samples from the same SFT model are often quite similar, making it hard and often very subjective to select a preferred sample. Second, we use a pool of human annotators that may have different preferences.  We emphasize that the disparity is not merely due to the crowd-workers; even when we (the paper's authors) annotated >200 examples, we observed just a 64% agreement rate.
>
> The agreement here estimates the frequency with which an annotator aligns with the majority of humans. Concretely, we collected 650 examples each annotated by 4 different crowdworkers (i.e., 2600 in total). For human-human agreement, each example was evaluated by one human and then cross-checked against the majority preference of the remaining three. The resulting agreement was then averaged over all four humans and 650 examples. For model-human agreement, we performed the same computation: comparing the model's response to the majority agreement of the 3 humans, then averaging over samples and human participants.
>
> We thank the reviewer for those questions, which we now expanded on in the paper.
>
> ----
>
> > *it would also be good to include different models for evaluating and even for simulating human feedback to see how much variance there is and increase the robustness of the results obtained with this platform.*
>
> We agree that understanding the robustness of the simulator to the oracle model is important. Figures 9, 10, and 11 in the appendices investigate the use of other models and prompts for simulation. Here is a table with a more in-depth analysis of the use of different simulated evaluators, their agreement with humans, their variance, and their biases. Our annotators are bolded. The last 4 columns in the following table refer to the metrics from figures 9  (bias, variance) and figure 11 (preference for list and for longer outputs) in appendices E.3. of the submitted manuscript.
>
> |  | Human agreement [%] | Price [$/1000 examples] | Time [seconds/1000 examples] | Bias | Variance | Proba. prefer longer | Proba. prefer lists |
> |---|---|---|---|---|---|---|---|
> | alpaca_eval_gpt4_fn | 71.0 | 14.5 | 5046 | 27.6 | 11.1 | 0.75 | 0.63 |
> | alpaca_farm_greedy_gpt4 | 66.4 | 15.3 | 878 | 30.2 | 19.3 | 0.60 | 0.59 |
> | humans | 65.7 | 300.0 | 36800 | 0.0 | 34.3 | 0.64 | 0.61 |
> | claude | 65.5 | 11.1 | 173 | 31.9 | 18.0 | 0.62 | 0.58 |
> | text_davinci_003 | 64.1 | 8.7 | 121 | 33.8 | 22.7 | 0.70 | 0.64 |
> | lmsys_gpt4 | 63.2 | 13.9 | 17982 | 34.7 | 16.1 | 0.74 | 0.64 |
> | alpaca_farm | 60.0 | 11.5 | 820 |  |  | 0.60 | 0.63 |
> | chatgpt_fn | 60.0 | 1.0 | 530 | 36.9 | 27.7 | 0.62 | 0.65 |
> | chatgpt | 57.2 | 0.8 | 285 | 39.4 | 34.1 | 0.59 | 0.56 |
> | cohere | 53.4 | 3.5 | 217 |  |  | 0.50 | 0.51 |
>
> ## Other
>
> > *It would also be good if the authors can include human evaluations in the paper.*
>
> All validation of our pipeline is done using human evaluations. Is the reviewer asking about releasing those annotations? If so, we have released them as part of our code release but have not linked to them due to the anonymity guidelines.

---

> > ### Comment · Reviewer_JtTg · 2023-08-11
> > **Response to Rebuttal**
> >
> > I thank the authors for their thorough responses and for going the extra mile to further improve the paper by running additional experiments and including an extensive discussion of its limitations. I also appreciate the emphasis of some details and results I might have missed in the Appendix, for using a wide range of models to perform evaluations, and for releasing human annotations.
> > I'm happy to say the rebuttal has addressed my main concerns.
> >
> > I believe this paper would be a very valuable contribution to the community by democratizing research on an important topic (RLHF) which has a relatively high entry barrier. I also find the experiments to be very thorough. In light of this, I have increased my score to 8 to reflect my strong support of this paper.

---

> > > ### Author Response · Authors · 2023-08-13
> > >
> > > Thank you for considering our rebuttal and for the encouraging words!

---

### Official Review · Reviewer_pSDY · 2023-07-05

**Soundness:** 3 good
**Presentation:** 4 excellent
**Contribution:** 3 good
**Rating:** 8
**Confidence:** 4

**Summary:**

The authors provide a simulator for experiments with LLMs that aim to learn from human-feedback, in particular, human binary comparisons. This allows researchers to run exploratory experiments with, e.g., RLHF, quickly and cheaply, without having to collect human data. The main contribution is the open source library that includes various components needed for these type of experiments. In the paper, the authors evaluate their simulator; using end-to-end validation, validating subparts of the library, and showcasing how it can be used to produce findings match those found when using human feedback.

**Strengths:**

Learning from human feedback is an important topic that has long been hard to study for any but the most well-resourced labs. Alpaca Farm will enable more research on this topic

Open source software that enables other researchers is always great.

The paper is well written.

Thorough validation.

The validation results, especially Figure 2, are impressive.

The finding in Figure 3 (reward model over-optimisation only happens with strong inter-rater disagreement) is intriguing (seems worth further study)


**Weaknesses:**

From more to less relevant concerns. I will raise my score if these are addressed sufficiently in the rebuttal.

The paper could explore the limitations of Alpaca Farm more. I don't actually believe that Llama-7B with some some instruction fine-tuning and a bit of PPO is comparable to GPT-3.5; you probably don't believe this either. However, Table 1 shows a winrate of 55% of "PPO". This is probably because your evaluation instructions are relatively easy, only single turn, you don't factor in adversarial inputs, and so on. This is fine, but it should be discussed more in the paper. In general, the paper would be better if it elucidated the limitations of your simulator a bit more, as there clearly are some. Try to break it, show not only where it works, but where it stops working.

There has been recent work that shows that MTurk crowdworkers use AI a lot. https://arxiv.org/abs/2306.07899 . So, it's not wonder that MTurk evaluations agree with the evaluations from you GPT evaluator models. How do you expect this to affect your results?

Figure 4: this experiment would be better if you used humans to evaluate the winrate on the demo. The x-axis is the evaluation in Alpaca Farm. The y-axis should be the "realistic evaluation", which means realistic data AND human evaluation.

Why does Figure 2 have 10 data points? What are these data points? Comparing to Table 1, it seems like you included GPT-4 on Figure 2; this seems misleading, as this only measures the evaluation part of your simulator, and not the training part. You should show a Figure where you only include models that were actually trained on the simulated preferences.

I have some worries thinking about that the training signal (pairwise comparisons) and the evaluators (pairwise comparisons) come from the same API-models. This might lead to overfitting to these particular evaluators. However, in practicse, this doesn't seem to be an issue (see Figure 2). Of course, if you work with humans, you may also use the same humans for training signal and eval (although best practice would probably be to use different humans)

Paper could be shorter, a bit much repetition.


-------
EDIT: some of these points have been convincingly addressed in the rebuttal, so I raise my score


**Questions:**

Why does Figure 2 have 10 data points? What are these data points? Comparing to Table 1, it seems like you included GPT-4 on Figure 2; this seems misleading, as this only measures the evaluation part of your simulator, and not the training part. You should show a Figure where you only include models that were actually trained on the simulated preferences.

There has been recent work that shows that MTurk crowdworkers use AI a lot. https://arxiv.org/abs/2306.07899 . So, it's not wonder that MTurk evaluations agree with the evaluations from you GPT evaluator models. How do you expect this to affect your results?


**Limitations:**

The paper could explore the limitations of Alpaca Farm more. I don't actually believe that Llama-7B with some some instruction fine-tuning and a bit of PPO is comparable to GPT-3.5; you probably don't believe this either. However, Table 1 shows a winrate of 55% of "PPO". This is probably because your evaluation instructions are relatively easy, only single turn, you don't factor in adversarial inputs, and so on. This is fine, but it should be discussed more in the paper. In general, the paper would be better if it elucidated the limitations of your simulator a bit more, as there clearly are some. Try to break it, show not only where it works, but where it stops working.

---

> ### Author Rebuttal · Authors · 2023-08-08
>
> We thank the reviewer for their review and insightful feedback, which we incorporated in the updated manuscript.
>
> ## Limitations
>
> >*The paper could explore the limitations of Alpaca Farm more.*
>
> We agree with the reviewer’s suggestion and have incorporated the feedback. Please see the [general rebuttal](https://openreview.net/forum?id=4hturzLcKX&referrer=%5BAuthor%20Console%5D(%2Fgroup%3Fid%3DNeurIPS.cc%2F2023%2FConference%2FAuthors%23your-submissions))
>
> ---
>
> > *I don't actually believe that Llama-7B with some instruction fine-tuning and a bit of PPO is comparable to GPT-3.5. [...] your evaluation instructions are relatively easy, only single turn, you don't factor in adversarial inputs, and so on*
>
> We agree with the reviewer’s concerns and have added all the necessary disclaimers for that result in the revised manuscript. Furthermore, we have detailed the mentioned limitations in a new section (refer to [general rebuttal](https://openreview.net/forum?id=4hturzLcKX&noteId=Hq0kpzSHSw)).
>
> Evaluation of instruction following models is an active area of research. Our team is actively working on that domain, and we intend to incorporate recent and upcoming evaluation improvements into AlpacaFarm.
>
> ---
>
> > *There has been recent work that shows that MTurk crowdworkers use AI a lot. https://arxiv.org/abs/2306.07899 [...]’How do you expect this to affect your results?*
>
> This is a significant potential concern that we were aware of. We undertook two measures to address this during the project:
> 1. We manually annotated over 200 examples without any language model and assessed our agreement with the auto annotator. The agreement was 64%, akin to the agreement between crowd-workers and the auto annotator and among different authors.
> 2. We continuously monitored the agreement between crowd-workers and GPT4, as well as the time taken for annotations. This monitoring led us to exclude 3 crowd-workers and their annotations, who were annotating particularly fast with a questionable agreement with GPT4.
>
> Though there might still be instances of LLM use we didn't detect, given these precautions, we're confident it hasn't majorly affected our results. We did not include a discussion of the paper since it was published after our submission, but we will include it in the updated manuscript.
>
> ---
>
> >*I have some worries thinking about that the training signal (pairwise comparisons) and the evaluators (pairwise comparisons) come from the same API-models. This might lead to overfitting to these particular evaluators. However, in practice, this doesn't seem to be an issue (see Figure 2).*
>
> We agree with everything that the reviewer said.  We now address this as a potential limitation, but as the reviewer pointed out, it doesn't appear to be a major concern in practice. This potential issue might be partly alleviated given that we utilize a diverse pool of annotators (simulated or crow-dworkers).
>
> ## Other
>
> > *You should show a Figure where you only include models that were actually trained on the simulated preferences.*
>
> We agree and have updated the figure in the main paper. We also clarified in Table 1 which models were only evaluated (and not trained) in both human and simulated settings.

---

> > ### Comment · Reviewer_pSDY · 2023-08-10
> > **Updated Figure**
> >
> > > We agree and have updated the figure in the main paper
> > Could I see the updated Figure please?

---

> > > ### Author Response · Authors · 2023-08-10
> > > **Link to figure & table**
> > >
> > > Please refer to this [anonymized link]( https://anonymfile.com/g3xjN/alpacafarm-graypoints.pdf) for a PDF containing the updated figure and table from the main manuscript. We apologize for the oversight in not providing any proof of our claim.

---

> > > > ### Comment · Reviewer_pSDY · 2023-08-10
> > > > **Response to figure**
> > > >
> > > > OK, this still looks good. Overall, you might benefit from experiments that train and evaluate a model with poor performance, so that the left side of the graph also has a blue dot.
> > > >
> > > > In total, the response has updated me enough to raise my score. It seems that I can't edit the original review right now, but I assume this will become available later.

---

> > > > > ### Author Response · Authors · 2023-08-13
> > > > >
> > > > > Thank you for considering the rebuttal and for your quick answer!

---

### Official Review · Reviewer_goj1 · 2023-07-07

**Soundness:** 4 excellent
**Presentation:** 4 excellent
**Contribution:** 4 excellent
**Rating:** 8
**Confidence:** 4

**Summary:**

The paper identifies three major challenges in training models with human feedback: (a) the cost of *preference* data collection, (b) the lack of trustworthy eval, and © the absence of implementations for reference methods. I completely agree with the fact that the process of training LLMs with human feedback is less understood due to the lack of published information and tools on them. The paper does a great job at addressing these dimensions in great detail with solid experimental results and thought-provoking findings. The paper establishes AlpacaFarm as a framework whose feedback and evaluation is synergetic with training on human feedback and human evaluation. Overall, the paper is well-written and clear to follow.


*Cost of preference data collection*

- The paper’s contribution in creating prompts for API LLMs that follow high human agreement and replicate human variability is very novel!
- I do not fully understand why the authors want the practitioners to collect high quality human feedback (post-AlpacaFarm simulations) if they already have AlpacaFarm API LLM annotators. It makes it difficult for me to digest how Alpaca Farm is reducing the cost of collecting preference data if eventually we do need to collect actual human feedback.
- I will be interested in understanding the gap between the performance of the methods trained with API LLM feedback vs human feedback by fixing the evaluator as Alpacafarm eval in the first case and humans in the other. Currently, the presented results focus on the gap between training with the simulated feedback + simulated evaluation and human feedback + human evaluation.

*Trustworthy eval*

- The experiments establish the high correlation between the simulated win-rates and human win-rates. It was interesting to observe that the rankings of the methods match well under the simulated environment and the real-world environment.
- I feel the paper lacks a discussion on evaluating the LLMs on the existing NLP datasets such as MMLU, SuperGLUE and benchmarks such as BigBench. I understand that 805 instructions may be a set of instructions that humans care about but I still feel that the prior works on creating datasets for model eval deserve credit in the main text.

*Reference Implementations*

- I agree that there are not a lot of implementations for reference methods, and really appreciate the authors for providing them.
- Given, PPO is known to be finicky and hard to stabilize, it would have been good to get some more details about the hyperparameter search in the main/supplementary material.
- Minor comment: It would be good to mention that Best-of-n policy uses n = 1024 more frequently in the plots.


**Strengths:**

Mentioned in the main comment.

**Weaknesses:**

No major issues with the contribution and experiments.

**Questions:**

More details on the reference method implementations in the main/appendix would be great.

**Limitations:**

No major issues.

---

> ### Author Rebuttal · Authors · 2023-08-08
>
> We thank the reviewer for their encouraging and thorough review. We will clarify and emphasize the answers to all their questions in the updated manuscripts.
>
> ## Training in simulating vs with humans
>
> > I will be interested in understanding the gap between the performance of the methods trained with API LLM feedback vs human feedback by fixing the evaluator as Alpacafarm eval in the first case and humans in the other.
>
> The main validation of the paper involves training and evaluation with the oracle LLM versus training and evaluation with humans. As the reviewer suggests, understanding the setting of training in simulation and evaluating with humans is also useful. We investigated this setting in Appendix B.2, where we show how to modify AlpacaFarm’s simulated annotators if one wants to use them as a source of supervision rather than as simulator.
>
> In light of the reviewer's feedback, we have now incorporated a summary of those results into the main text. In essence, the PPO model trained on AlpacaFarm's simulator achieved a 43% human-evaluated win-rate, which is significantly worse than the 55% win-rate of the PPO model trained on human feedback. However, we show that the PPO model trained using feedback from a low-variance GPT4 annotator achieves a 50% win-rate.
>
> ---
>
> > I do not fully understand why the authors want the practitioners to collect high-quality human feedback (post-AlpacaFarm simulations) if they already have AlpacaFarm API LLM annotators. [...] how Alpaca Farm is reducing the cost of collecting preference data [...]
>
> We thank the reviewer for pointing this out and will make the distinction clearer in the manuscript. The primary cost-saving benefit of introducing a simulator before human engagement is to facilitate an affordable method development, e.g., when developing a new RLHF algorithm. By refining the model pipeline in simulation (often entailing multiple rounds of training), we need only run the final training with human feedback.
>
> As discussed above, Appendix B.2 shows how to repurpose AlpacaFarm as a source of supervision that does not require a final human round, further decreasing the cost. However, we find that PPO trained with this automated source of supervision still lag behind human PPO. We see narrowing this gap as a promising direction for future work.
>
> ## Other
>
> > I feel the paper lacks a discussion on evaluating the LLMs on the existing NLP datasets such as MMLU, SuperGLUE and benchmarks such as BigBench.
>
>  >get some more details about the hyperparameter search in the main/supplementary material.
>
> > mention that Best-of-n policy uses n = 1024 more frequently.
>
> We thank the reviewer for their feedback, which we have incorporated in the updated manuscript.

---

> > ### Comment · Reviewer_goj1 · 2023-08-12
> > **Response to Rebuttal**
> >
> > Hi, I thank the authors for rebuttal.
> >
> > - It is interesting to see that the optimal LLM feedback setups are different if you want to use them for supervision versus simulators.
> >
> > On "affordable method development with AlpacaFarm as a simulator":
> > - I believe that the AlpacaFarm's "cost-effectiveness" would lie in their faithful automatic evaluation that achieves high agreement with the human judgments, instead of AlpacaFarm's feedback data itself.
> > - Let us say, a practitioner has N plausible RL algorithms for aligning their LLM. Without any doubt, collecting feedback data (pairwise judgments) from AlpacaFarm will be cheaper than human feedback.
> > - **Scenario 1**: Under the cost-effective argument in the paper, the practitioner would test their RL algorithms on AlpacaFarm feedback data, and use the best method (on AlpacaFarm eval) to train on human feedback data. Finally, they would perform a human eval of the algorithm trained with human feedback.
> > - **Scenario 2**: The practitioners could collect human feedback data and select the algorithm that works best on AlpacaFarm eval. Finally, they can perform a human eval of the best algorithm thus found.
> > - In my opinion, **Scenario 2** is more straightforward and cheaper than **Scenario 1**.
> >
> > Overall, I am satisfied with the paper's findings and experiments. Good luck to the authors. I will keep my scores unchanged since there were not any major concerns in the original rebuttal anyway.

---

> > > ### Author Response · Authors · 2023-08-13
> > >
> > > Thank you for the detailed answer. This is a great point! We agree that Scenario 2 is cheaper in the case of a single-round RLHF, which is the experimental setting we consider!
> > >
> > > In the case of multiple RLHF rounds, scenario 2 would likely become more expensive as a different set of human preferences would need to be collected for every considered model! We hope that future work will consider AlpacaFarm in multi-round settings.

---

### Author Rebuttal · Authors · 2023-08-08

# General

We thank the reviewers for their insightful and constructive feedback.

We are pleased that the reviewers found our paper well-written [goj1, pSDY, JtTg, NPeD], thorough [goj1, pSDY], and believe that it may be an impactful and valuable contribution to the community [goj1, pSDY, JtTg, NPeD].

There are two questions/feedback shared by different reviewers: the need for a limitation section [pSDY, JtTg, NPeD] and how we came up with 25% label noise [JtTg, NPeD]. We address general feedback here and will answer questions specific to each reviewer separately.

We will upload the updated manuscript as soon as OpenReview allows us to. We appreciate the reviewers' feedback, which we believe has improved the quality of our work.

## Limitation section

We agree with the reviewers' suggestions about highlighting our limitations [pSDY, JtTg, NPeD]. We have taken advantage of the additional page in the updated manuscript to include a limitation section, which consolidates and elaborates on the limitations discussed throughout the paper and appendices. We briefly highlight some of these here:

- **Validation**: Although we provided strong validation results of AlpacaFarm in Section 4, there are limitations in how we perform such validation. First, instructions are relatively simple and are single-turn (even those from the real world-demo). Second, we only consider LLaMA 7B as the base model as this was the only model powerful enough for learning from human feedback at the time of submission. Finally, human validation is based on feedback from 13 crowd-workers, which may not reflect broader human preferences.
- **Assumption**: We assume access to an “oracle” LLM, which is more powerful than the ones we are training. While this may be true in research settings, it's not always the case in practice.
- **Generalization of Hyperparameters**: Specific hyperparameters, like the KL regularization coefficient, might not translate seamlessly from simulated environments to training with human preferences. The AlpacaFarm simulator is thus likely more useful for method development than hyperparameter tuning (and we only validated it in that setting).

## 25% noise

In response to reviewers [JtTg, NPeD]'s question about how we arrived at a 25% label flipping rate, we have highlighted and expanded our discussion about it in the updated manuscript.

In summary, we selected 25% based on two factors: estimated human (intra- and inter-annotator) variability (\~0.35) and overoptimization. Injecting a 25% label flip brings our annotator variability closer to the human one (\~0.43), and it was also the point at which we began to observe overoptimization (see rebuttal figure).

For a more in-depth discussion about annotator variability, including the variability of standard simulated annotators (<0.1), please refer to Appendix B.1 of the original manuscript.

---

### Comment · Area_Chair_7zSo · 2023-08-17
**Questions from area chair**

Hi authors,

To facilitate discussion about the paper, it would be great if you could answer the following question.

You use a constant label flip rate of 25%. This seems to ignore the fact that different prompts x will have different levels of human ranker variability. For example preferences about prompts that relate to taste ("Is rock better than jazz etc.") will be highly variable, whereas math problems ("does 2+2 =4?") will have zero variability in responses. Can you comment on this? I am not saying this is a huge problem, but wanted to have your opinion.

---

> ### Author Response · Authors · 2023-08-18
>
> Thank you for your insightful question. We recognize and agree that the degree of randomness depends on the example. Although the constant label flip rate of 25% doesn't directly address this variability, we have incorporated two mechanisms to simulate such example-specific randomness:
>
> 1. **Sampling the annotator**: For each annotation, we sample a different simulated annotator. This strategy is designed to capture the variability between annotators, especially for more subjective queries like "Is rock better than jazz?".
>
> 2. **Sampling from each annotator**: In some examples (e.g. Question: "Is 2+2=4?", Answer 1: "Yes", Answer 2: "yes") the choice of the preferred answer possess inherent randomness even for a given annotator. In such instances, our simulated annotators have two sources randomness. Firstly, we sample responses from all annotators with a temperature of 1.0. Secondly, we randomize the order in which answers appear in the prompt. Both of these methods should not influence the annotator's preference if that preference is clear. However, in scenarios where preferences are ambiguous, they lead to high randomness.
>
> The constant label flip rate of 25% acts as an example-agnostic randomness source. While it might seem excessive for questions with objective answers (e.g., Q:"does 2+2 =4?", A1: "Yes", A2: "No"), our experimental observations suggest this rate is necessary to emulate human variability (refer to our [general rebuttal](https://openreview.net/forum?id=4hturzLcKX&noteId=Hq0kpzSHSw)). Qualitatively, we have also found human annotations to exhibit high variability even when there was an objectively better answer. This might be attributed to the typically lengthy responses which annotators might not scrutinize rigorously, thereby overlooking occasional errors.
>
> We acknowledge that our method of emulating human variability has many limitations, which we discuss in the new limitation section. We hope that it will serve as a preliminary step in sparking the desired discourse and are eager to see how the community further refines this approach!

---

### Decision · Program_Chairs · 2023-09-21

**Decision:**

Accept (spotlight)

**Comment:**

This paper starts from a simple but compelling premise (make a simulator to replace humans in RLHF). There is massive potential for impact since RLHF research needs a cheap way to test-drive algorithms before they are deployed on real (and expensive) humans.  Ethical concerns seem adequately addressed.

There are some inevitable modelling simplifications (like the constant rate of flipping label), but no significant faults.